# Reference-Guided Identity Preserving Face Restoration

**Mo Zhou**[*]                                                 *cdluminate@gmail.com*
*Google & Johns Hopkins University*

**Keren Ye**                                                   *yek@google.com*
*Google*

**Viraj Shah**                                                 *virajshah@google.com*
*Google*

**Kangfu Mei**                                                 *kangfumei@google.com*
*Google*

**Mauricio Delbracio**                                         *mdelbra@google.com*
*Google*

**Peyman Milanfar**                                            *milanfar@google.com*
*Google*

**Vishal M. Patel**                                            *vpatel36@jhu.edu*
*Johns Hopkins University*

**Hossein Talebi**                                             *htalebi@google.com*
*Google*

**Reviewed on OpenReview:** *https://openreview.net/forum?id=g9YzUDnUUS*

## Abstract

Preserving face identity is a critical yet persistent challenge in diffusion-based image restoration. While reference faces offer a path forward, existing methods typically suffer from partial reference information and inefficient identity losses. This paper introduces a novel approach that directly solves both issues, involving three key contributions: 1) Composite Context, a representation that fuses high- and low-level facial information to provide ensembled guidance than traditional singular representations, 2) Hard Example Identity Loss, a novel loss function that uses the reference face to address the identity learning inefficiencies of the standard identity loss, 3) Training-free multi-reference inference, a new method that leverages multiple references for restoration, despite being trained with only a single reference. The proposed method demonstrably restores high-quality faces and achieves state-of-the-art identity preserving restoration on benchmarks such as FFHQ-Ref and CelebA-Ref-Test, consistently outperforming previous work.

## 1 Introduction

Recently, image restoration (Wang et al., 2018; 2024b; Yu et al., 2024; Lin et al., 2024; Wu et al., 2024; Yang et al., 2023b) has seen significant improvements along with the rise of diffusion models (Ho et al., 2020; Song et al., 2021a), particularly in terms of generated image quality (Rombach et al., 2022; Podell et al., 2024). However, the state-of-the-art restoration methods, including the face-specific ones (Zhou et al., 2022; Lin et al., 2024; Hsiao et al., 2024; Ying et al., 2024), still suffer from unsatisfactory identity preservation when

---

[*]Work done during internship at Google LLC.

processing facial imagery. This limitation can substantially degrade the user experience, given the human perceptual acuity for subtle variations in facial features.

In some real-world applications, such as digital albums, when restoring a low-quality face image, it is possible to leverage other high-quality images from the same person as references to better preserve the identity and appearance. Consequently, many reference-based face restoration methods (Hsiao et al., 2024; Ying et al., 2024; Zhang et al., 2025; Li et al., 2022) have been proposed. These efforts involve designing novel architectures for reference face conditioning (Ying et al., 2024; Hsiao et al., 2024), formulating loss functions for identity preservation and image quality (Hsiao et al., 2024; Zhang et al., 2025), and curating specialized reference-based face restoration datasets (Hsiao et al., 2024; Li et al., 2022). Nevertheless, the existing methods do not fully exploit the potential of reference faces – only partial representation of the reference face is used, and not sufficiently involved in supervision, and hence there is room for improvement in both identity preservation and image quality.

In this paper, to simplify architecture and more effectively utilize the reference face images and further enhance the performance of reference-based face restoration, we propose two independent modules that exploit the reference face in two different aspects: representation and supervision. Respectively, we propose Composite Context for the representation aspect, and Hard Example Identity Loss for the supervision aspect. They will be explained in the following text.

First, we propose *Composite Context*, an ensembled representation for the reference face. It is designed in order to avoid using only a partial representation of the reference face, as related works only use either the high-level information (such as person identity), or the low-level appearance information (such as skin texture).[1] It consists of multiple pre-trained face representations that focus on different information in the reference face from high-level to low-level. Specifically, it includes identity embedding (Deng et al., 2022) as high-level identity information; and general face representation (Zheng et al., 2022) comprising both high-level semantic information and low-level face information. In contrast, prior methods (Ying et al., 2024; Wang et al., 2025; Hsiao et al., 2024) rely on a single feature type, creating an information bottleneck that forces the model to restore a face with only partial guidance – either high-level identity or low-level appearance, but never both. Our Composite Context is conceptually similar to multi-modal approaches in generation (Podell et al., 2024; Mei et al., 2025) but is the first to combine specialized face encoders in this manner for restoration, moving beyond the information bottleneck.

Second, we propose *Hard Example Identity Loss*. It is designed in order to avoid the learning inefficiency issue with the existing loss functions.[2] It is a simple yet effective extension of the existing identity loss (Hsiao et al., 2024), motivated by the empirical observation that traditional identity loss suffers from learning inefficiencies – a well-known issue in metric learning (Schroff et al., 2015; Musgrave et al., 2020; Roth et al., 2020). While hard example mining is a known technique in metric learning, we are the first to identify and solve this specific learning inefficiency in face restoration. Our Hard Example Identity Loss offers a novel and targeted improvement that resolves this long-overlooked issue. In particular, the ground-truth faces are not hard enough (see "Triplet Selection" in (Schroff et al., 2015) for the meaning of "hard example"), which makes the identity loss magnitude very small after a short period of training. By simply incorporating a hard sample, namely the reference face, into the identity loss, the learning inefficiency problem can be effectively addressed, and hence leads to a significant performance improvement. In contrast, all previous works (Wang et al., 2025; Hsiao et al., 2024; Zhang et al., 2025) overlooked this issue.

Apart from the representation and supervision aspects, while our method is designed to take a single reference face image during training, it can support multiple reference face images through a simple method based on classifier-free guidance (Ho & Salimans, 2022) at the inference stage, which requires no extra training. Such design is more scalable due to multi-reference training data scarcity.

Our qualitative and quantitative results on the FFHQ-Ref (Hsiao et al., 2024) and CelebA-Ref-Test (Hsiao et al., 2024) datasets demonstrate the effectiveness of our method. Though simple, our method performs competitively compared to the previous methods, especially in face identity preservation.

---

[1]The details are discussed in Section 3.1.
[2]See Section 3.2 for detailed explanations including the loss curves.

**Contributions**. Our contributions are threefold regarding reference-based face image restoration:

- We introduce "Composite Context", an ensembled face representation that integrates multi-level information from a reference face to enable more effective guided restoration.
- We propose "Hard Example Identity Loss", a novel variant of the standard identity loss that incorporates the reference face to improve learning efficiency and identity preservation.
- Our model can leverage multiple references for restoration, despite being trained with only a single reference. This approach eliminates the need for multi-reference training datasets, which are difficult and costly to curate at scale.

## 2 Related Work

**Image Restoration.** As diffusion models (Ho et al., 2020; Rombach et al., 2022; Song et al., 2021a; 2023; Dhariwal & Nichol, 2021; Song et al., 2021b) gain popularity in image generation, LDM (Rombach et al., 2022) has recently become a popular backbone for general image restoration (Wang et al., 2024b; Lin et al., 2024; Yu et al., 2024; Yang et al., 2023b; Wu et al., 2024; Mei et al., 2025). However, humans are perceptually highly sensitive to subtle differences in face images, general image restoration techniques typically perform poorly, especially in terms of identity preservation and maintaining face image realism. In this case, face-specific restoration models are preferred.

**No-reference Face Restoration.** When there is no reference face, generative models can be used to hallucinate details while restoring a degraded facial image (Zhou et al., 2022; Lin et al., 2024; Wang et al., 2025; Chen et al., 2018; Li et al., 2020b; Wang et al., 2021a; Yang et al., 2023a; Li et al., 2020a). CodeFormer (Zhou et al., 2022) presents a Transformer (Vaswani et al., 2017) to model the global composition and context of the low-quality faces for code prediction, enabling the generation of natural faces that closely approximate the target faces. DiffBIR (Lin et al., 2024) presents a two-stage pipeline for blind face restoration, involving the degradation removal and information regeneration. OSDFace (Wang et al., 2025) proposes a visual representation embedder to capture information from low-quality face and incorporate the face identity loss for identity preservation. A common challenge in no-reference face restoration is identity preservation as no additional information is provided.

**Reference-based Face Restoration**. High-quality reference face images, when available, can help identity preservation when restoring a low-quality face of the same person (Min et al., 2024; Hsiao et al., 2024; Ying et al., 2024; Zhang et al., 2025; Li et al., 2022; Varanka et al., 2024). DMDNet (Li et al., 2022) proposes a dual memory dictionary for both general and identity-specific features for blind face restoration. RestorerID (Ying et al., 2024) presents a Face ID Adapter and incorporates the identity embedding of the reference face as a tuning-free face restoration method. InstantRestore (Zhang et al., 2025) leverages a one-step diffusion model, and proposes a landmark attention loss to enhance identity preservation. RefLDM (Hsiao et al., 2024) incorporates the CacheKV mechanism and a timestep-scaled identity loss into an LDM (Rombach et al., 2022) to effectively utilize multiple reference faces. However, methods like RefLDM require multiple reference images during training, which presents a data scalability challenge. We address this by proposing a more practical paradigm of training with a single reference while effectively supporting multiple references at inference time. Personalization methods (Varanka et al., 2024; Liu et al., 2025) utilizes reference faces with the goal of customizing the model for individual users.

## 3 Our Approach

Given a low-quality (LQ) face image $x_{LQ}$, and a high-quality (HQ) reference face image $x_{REF}$ from the same person, we aim to restore the LQ image while preserving the person identity by leveraging the reference face. The resulting image should be close to the ground truth HQ image $x_{HQ}$ in terms of both identity similarity and perceptual similarity.

To this end, we adopt a general LDM (Rombach et al., 2022) backbone pretrained for text-to-image synthesis. Following the previous works (Rombach et al., 2022; Hsiao et al., 2024), we incorporate the LQ input image

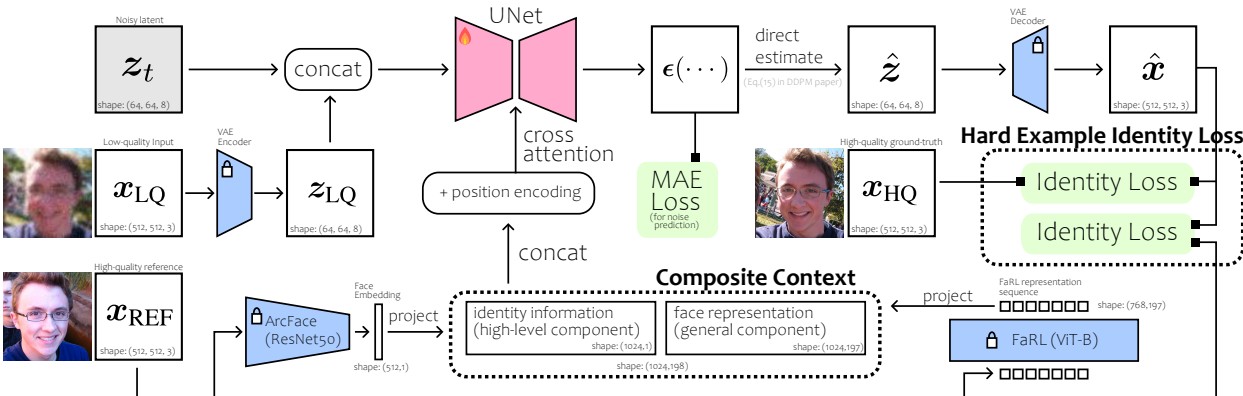

Figure 1: Overview of our method. The Composite Context and Hard Example Identity Loss are designed for fully exploiting the reference face and hence better identity preservation. The $z_t$ is noisy latent, $x_{\text{LQ}}$ is low-quality face image input ($z_{\text{LQ}}$ is its corresponding VAE latent), $x_{\text{REF}}$ is high-quality reference face, $x_{\text{HQ}}$ is high-quality ground-truth face image, $\hat{z}$ is the direct estimate of the denoised result (*i.e.*, Eq. (15) in DDPM (Ho et al., 2020)), and $\hat{x}$ is the VAE decoded direct estimate. All pre-trained modules are frozen. The UNet (Ronneberger et al., 2015) and projection matrices for Composite Context are trained. The total loss includes MAE loss and Hard Example Identity Loss.

$x_{\text{LQ}}$ by conditioning the diffusion model on its corresponding VAE latent $z_{\text{LQ}}$ through concatenating it to the noise latent $z_t$. In this way, the model $\epsilon(z_t, z_{\text{LQ}}, t)$ can serve as a fundamental face image restoration model.

In order to comprehensively leverage the reference face for better identity preservation, we propose two independent modules: Composite Context (CC) and Hard Example Identity Loss (HID), which will be detailed in Section. 3.1 and Section. 3.2 below. In brief, the Composite Context is an ensembled representation $c$ from the reference face $x_{\text{REF}}$. It is used as a condition for $\epsilon(z_t, z_{\text{LQ}}, c, t)$ through cross-attention mechanism (Rombach et al., 2022). The Hard Example Identity Loss $\mathcal{L}_{\text{HID}}$ will take advantage of the reference to enhance identity preservation. See Figure 1 for the overview.

### 3.1 Composite Context for Ensembled Reference Face Representation

Different from no-reference face restoration methods, reference-based methods (Hsiao et al., 2024; Ying et al., 2024; Zhang et al., 2025) assume that a high-quality reference face from the same person is available. To thoroughly leverage this advantage, we propose Composite Context, an ensembled representation of the reference face image that covers multi-level information from the reference face, including high-level semantic information (such as person identity) and low-level appearance information (such as skin texture). Unlike previous works (Wang et al., 2025; Hsiao et al., 2024; Ying et al., 2024; Zhang et al., 2025) that only leverage partial information from the reference face through a single representation, Composite Context allows the model to comprehensively leverage the reference face at different levels. Therefore, Composite Context may benefit identity preservation.

Given a reference face image $x_{\text{REF}}$ which belongs to the same identity as $x_{\text{LQ}}$, we can leverage a collection of pre-trained face representation models for various purposes to extract the respective representations, and combine them together as a vector sequence. In particular, Composite Context consists of the following multi-level components:

- **High-level features:** ArcFace (Deng et al., 2022) embedding representing person identity. It is a face recognition model which enforces an angular margin in its embedding space. We assume that $\phi_{\text{H}}(\cdot)$ is the pre-trained ArcFace model in the standard ResNet50 He et al. (2016) architecture, and $W_{\text{H}}$ is a projection matrix from the dimensionality of face embedding to the dimension of UNet cross-attention. The projected embedding $W_{\text{H}}\phi_{\text{H}}(x_{\text{REF}})$ is the first part of the Composite Context.

- **General features:** FaRL (Zheng et al., 2022) representation representing various high-level semantic (*e.g.*, face attributes) and low-level information (*e.g.*, visual appearance) of the reference face. FaRL is a general face representation model learned in a visual-linguistic manner, with image-text contrastive learning and masked image modeling simultaneously (Zheng et al., 2022). We assume that $\phi_{\mathrm{G}}(\cdot)$ is a pre-trained FaRL model (ViT-B (Dosovitskiy et al., 2021) architecture), and $\boldsymbol{W}_{\mathrm{G}}$ is the projection matrix to the dimension of UNet cross-attention. We use the whole output sequence (197 tokens) from FaRL to maximize reference face utility. The projected sequence $\boldsymbol{W}_{\mathrm{G}}\phi_{\mathrm{G}}(\boldsymbol{x}_{\mathrm{REF}})$ is the second part of the Composite Context.

After obtaining those representations from the reference face $\boldsymbol{x}_{\mathrm{REF}}$, they are concatenated, and added with the standard sinusoidal positional encoding (Vaswani et al., 2017) as the Composite Context:

$$\boldsymbol{c} = \mathrm{Concat}\big[\boldsymbol{W}_{\mathrm{H}}\phi_{\mathrm{H}}(\boldsymbol{x}_{\mathrm{REF}}), \boldsymbol{W}_{\mathrm{G}}\phi_{\mathrm{G}}(\boldsymbol{x}_{\mathrm{REF}})\big] + \boldsymbol{e}_{\mathrm{position}}, \tag{1}$$

where $\boldsymbol{e}_{\mathrm{position}}$ denotes sinusoidal positional encoding (Vaswani et al., 2017). Since all the Composite Context components are from pre-trained models, the sequence length is fixed at $1 + 197 = 198$ for any reference face. Finally, the Composite Context $\boldsymbol{c}$ is incorporated into the model through the cross-attention conditioning mechanism (Rombach et al., 2022) as $\boldsymbol{\varepsilon}(\boldsymbol{z}_t, \boldsymbol{z}_{\mathrm{LQ}}, \boldsymbol{c}, t)$. See Figure 1 for the overall diagram of the proposed method.

## 3.2 Hard Example Identity Loss for Improved Learning Efficiency

One of the goals of face restoration is to preserve the identity, which means the restored face should match the identity of the HQ image. To achieve this, many recent works (Hsiao et al., 2024; Wang et al., 2025; Zhang et al., 2025) incorporate the identity loss, which is based on a pre-trained face embedding model (Deng et al., 2022; Schroff et al., 2015) such as ArcFace (Deng et al., 2022). In particular, RefLDM (Hsiao et al., 2024) presents a timestep-scaled identity loss $\mathcal{L}_{\mathrm{ID}}$ as:

$$\mathcal{L}_{\mathrm{ID}}(\boldsymbol{x}_{\mathrm{HQ}}, \hat{\boldsymbol{x}}) = \sqrt{\bar{\alpha}_t} \cdot \big(1 - \cos\langle\phi_{\mathrm{H}}(\boldsymbol{x}_{\mathrm{HQ}}), \phi_{\mathrm{H}}(\hat{\boldsymbol{x}})\rangle\big), \tag{2}$$

where $\phi_{\mathrm{H}}$ denotes the face embedding model Deng et al. (2022), the notation $\sqrt{\bar{\alpha}_t}$ is inherited from DDPM (Ho et al., 2020), and $\hat{\boldsymbol{x}}$ is the direct estimate of $\boldsymbol{x}_0$ at time step $t$, *i.e.*, Eq. (15) in DDPM (Ho et al., 2020). The time-step scaling factor $\sqrt{\bar{\alpha}_t}$ mitigates the out-of-domain behavior of the identity loss at a very noisy step $t$, and emphasizes identity preservation at less noisy steps. However, a learning inefficiency issue is overlooked.

During experiments, we observe that the identity loss in Eq. (2) decreases quickly and plateaus at a very small magnitude, as shown by the blue curve in Figure 2. In the metric learning literature (Schroff et al., 2015; Musgrave et al., 2020; Roth et al., 2020; Zhou & Patel, 2022; Zhou et al., 2024), there is a similar phenomenon where the loss value is small when the training samples are not hard enough (see "Triplet Selection" in (Schroff et al., 2015)), which usually leads to poor generalization. Their countermeasure is to mine some hard examples (Schroff et al., 2015) that can trigger a larger loss value so the model performance can be drastically influenced (Roth et al., 2020). Inspired by such solution to the learning inefficiency issue, we propose to leverage the reference face $\boldsymbol{x}_{\mathrm{REF}}$ as a hard example in addition to $\boldsymbol{x}_{\mathrm{HQ}}$. Based on this, we design a simple extension to the identity loss $\mathcal{L}_{\mathrm{ID}}$ as the "Hard Example Identity Loss" incorporating the hard example, namely the reference face.

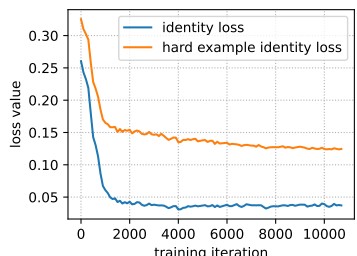

Figure 2: Loss curves of $\mathcal{L}_{\mathrm{ID}}$ and $\mathcal{L}_{\mathrm{HID}}$ during the training process. The curves are truncated to the beginning part of the training process.

Let $\lambda$ be a hyper-parameter for balancing the influence of $\boldsymbol{x}_{\mathrm{HQ}}$ and $\boldsymbol{x}_{\mathrm{REF}}$ during training. Formally, the Hard Example Identity Loss $\mathcal{L}_{\mathrm{HID}}$ is also based on the direct estimate $\hat{\boldsymbol{x}}$, and is defined as:

$$\mathcal{L}_{\mathrm{HID}}(\boldsymbol{x}_{\mathrm{HQ}}, \boldsymbol{x}_{\mathrm{REF}}, \hat{\boldsymbol{x}}) = (1 - \lambda)\mathcal{L}_{\mathrm{ID}}(\boldsymbol{x}_{\mathrm{HQ}}, \hat{\boldsymbol{x}}) + \lambda\mathcal{L}_{\mathrm{ID}}(\boldsymbol{x}_{\mathrm{REF}}, \hat{\boldsymbol{x}}). \tag{3}$$

As shown by the orange curve in Figure 2, our Hard Example Identity Loss will no longer plateau at a very small value because a "harder" example is introduced, and hence will alleviate the learning inefficiency issue.

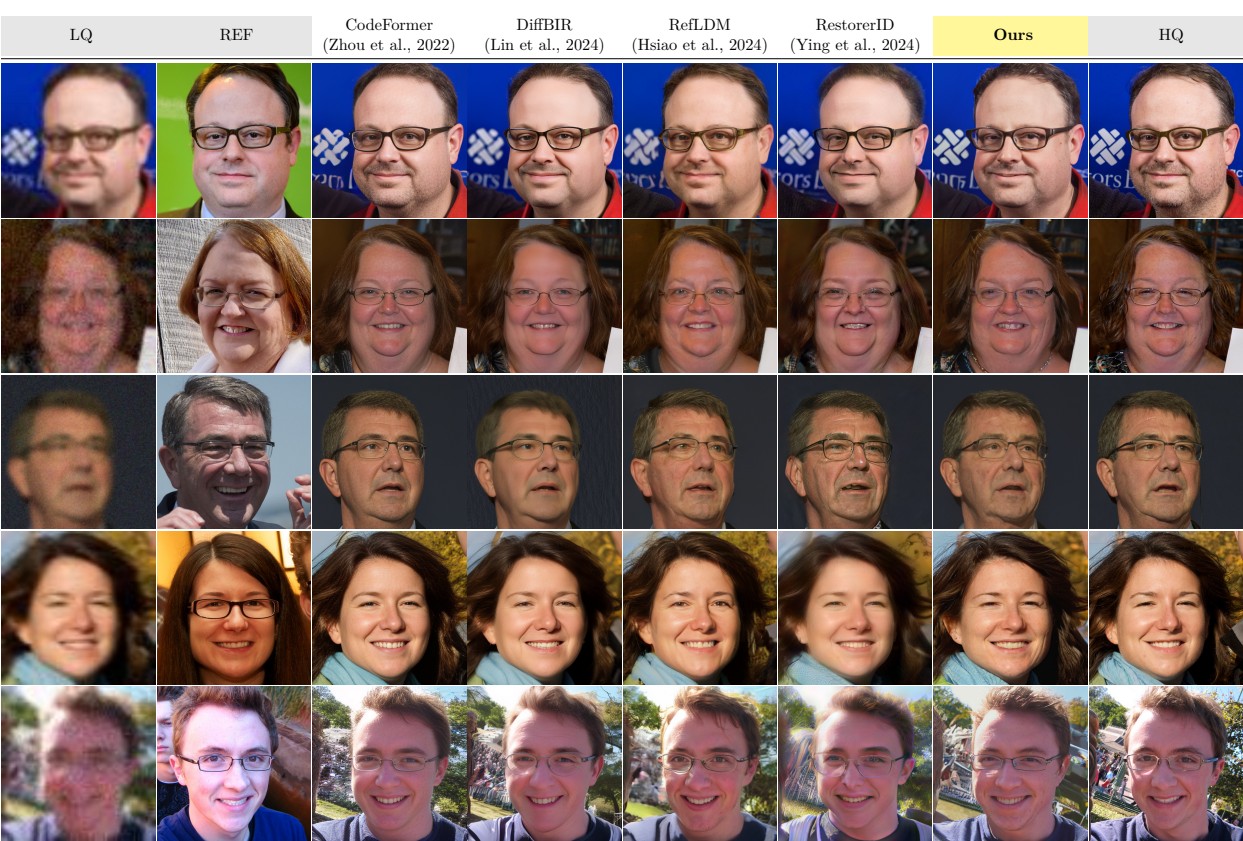

Figure 3: Qualitative comparison with other state-of-the-art face restoration methods on FFHQ-Ref Moderate (Hsiao et al., 2024) test set. The "REF" column is the reference face. Zoom in for details.

While simple in its form, the introduction of the reference face is very effective and can clearly improve the identity preservation. As a different interpretation of the introduction of the reference face, it is noted that the input faces are noisy (as they are direct estimations during DDPM), which inherently makes the face embedding and the identity loss noisy. In this case, introducing the additional contrastiveness through the reference face can potentially lead to a regularization effect, stabilizing the gradients from the identity loss. The total loss of our model is the L-1 diffusion loss (*aka.* MAE) and the Hard Example Identity Loss with a balancing hyper-parameter $w_{\mathrm{HID}}$:

$$\mathcal{L}_{\mathrm{total}} = \mathcal{L}_{\mathrm{MAE}} + w_{\mathrm{HID}} \cdot \mathcal{L}_{\mathrm{HID}}. \tag{4}$$

### 3.3 Training-Free Extension for Multi-Reference Faces

Classifier-free guidance (Ho & Salimans, 2022) is an effective technique for improving diffusion model performance, which is also widely adopted in the image restoration literature (Lin et al., 2024; Yu et al., 2024; Wang et al., 2024b). Since our model involves both the LQ condition $z_{\mathrm{LQ}}$ and $c$, we follow (Brooks et al., 2023) for their classifier-free guidance formulation:

$$\tilde{\epsilon}(z_t, z_{\mathrm{LQ}}, c, t) = (1 - s_i)\epsilon(z_t, \varnothing, \varnothing, t) + (s_i - s_c)\epsilon(z_t, z_{\mathrm{LQ}}, \varnothing, t) + s_c\epsilon(z_t, z_{\mathrm{LQ}}, c, t), \tag{5}$$

where $s_c$ controls the guidance effect of the composite context $c$, and $s_i$ controls the guidance effect of the LQ latent $z_{\mathrm{LQ}}$. The two hyper-parameters $s_i$ and $s_c$ can be adjusted at the inference stage.

While our method is designed to take only one reference face image, it can be extended to support multiple reference faces through a simple ensemble. Let $C = \{c_i\}_{i=1,...,N}$ be a set of composite contexts obtained

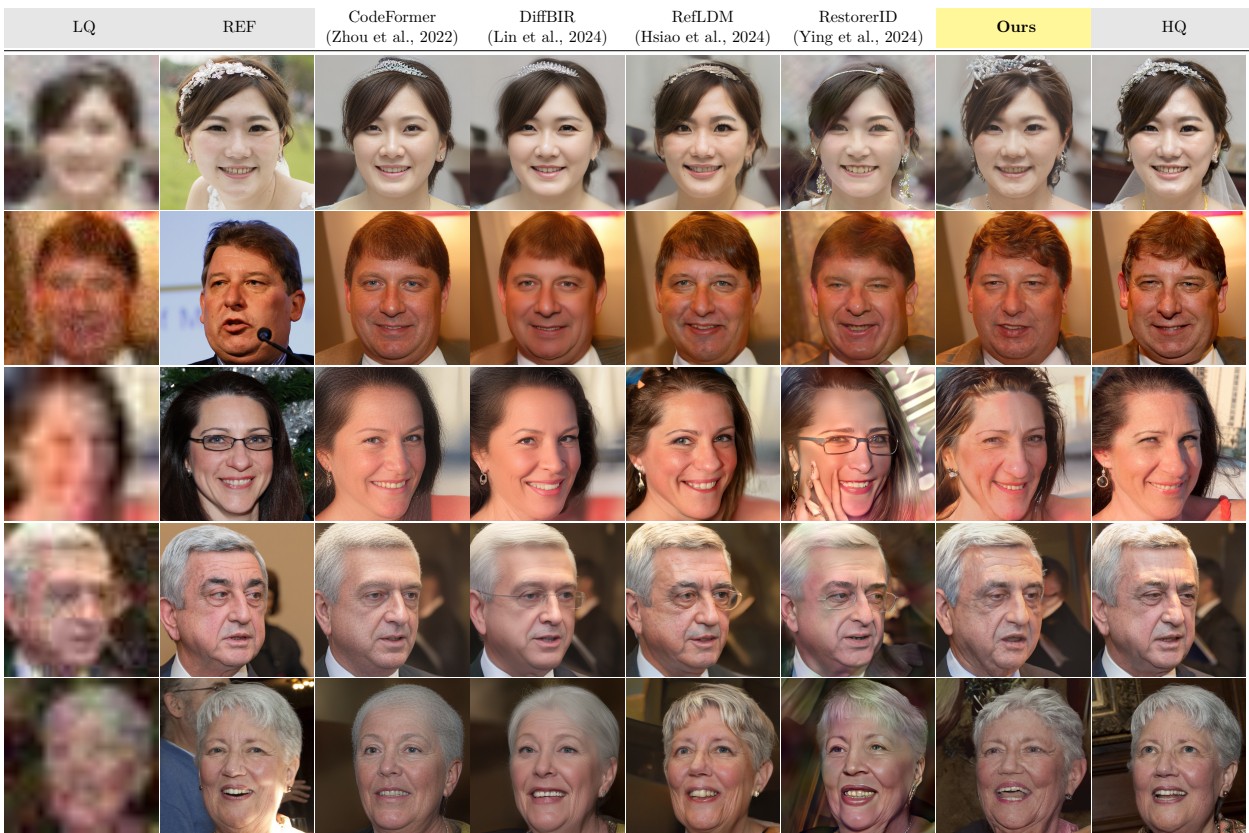

Figure 4: Qualitative comparison with other state-of-the-art face restoration methods on FFHQ-Ref Severe (Hsiao et al., 2024) test set. The "REF" column is the high-quality reference face image.

from $N$ reference face images. The multi-reference inference is formulated as:

$$\tilde{\boldsymbol{\epsilon}}(\boldsymbol{z}_t, \boldsymbol{z}_{\mathrm{LQ}}, \boldsymbol{C}, t) = (1 - s_i)\boldsymbol{\epsilon}(\boldsymbol{z}_t, \varnothing, \varnothing, t) + (s_i - s_c)\boldsymbol{\epsilon}(\boldsymbol{z}_t, \boldsymbol{z}_{\mathrm{LQ}}, \varnothing, t) + \frac{s_c}{N}\sum_{i=1}^{N}\boldsymbol{\epsilon}(\boldsymbol{z}_t, \boldsymbol{z}_{\mathrm{LQ}}, \boldsymbol{c}_i, t). \quad (6)$$

Inspired by (Hsiao et al., 2024), the identity is expected to be better preserved when more reference faces are provided. Different from (Hsiao et al., 2024) which uses multiple reference faces for training, our method only requires one reference face during training while being able to use multiple reference faces during inference. Our method alleviates the data scarcity issue in the multi-reference face scenario, where most training samples only have a single reference face (Hsiao et al., 2024). Such paradigm could be quite scalable in terms of the amount of reference face training data.

## 4 Experiments

**Datasets.** Our model is trained on the FFHQ-Ref (Hsiao et al., 2024) dataset, which is a subset of FFHQ (Karras et al., 2019) by person identity clustering. It comprises 18816 images for training and 857 images for testing. We follow (Wang et al., 2021b) for their second-order degradation simulation pipeline during training. For training data augmentation, we use random horizontal flipping with 0.5 probability, and random color jittering with 0.5 probability. For testing purposes, we adopt the identical test data from (Hsiao et al., 2024), namely FFHQ-Ref Moderate, FFHQ-Ref Severe, and CelebA-Ref-Test (Hsiao et al., 2024). In this paper, the face image resolution is always $512 \times 512$ following previous works (Zhou et al., 2022; Lin et al., 2024; Hsiao et al., 2024; Ying et al., 2024). Note, while most previous works do not use identical training data and may potentially suffer from test data leakage (Hsiao et al., 2024), our training and test images are completely identical to RefLDM (Hsiao et al., 2024) (NeurIPS'24) for a fair comparison.

Table 1: Comparison with state-of-the-art face restoration methods on FFHQ-Ref Moderate and Severe (Hsiao et al., 2024). The "#REF" means the number of reference face used. More details including standard deviation are available in Appendix A.3.1.

| Method | #REF | FFHQ-Ref Moderate | | | | | | | FFHQ-Ref Severe | | | | | | |
|---|---|---|---|---|---|---|---|---|---|---|---|---|---|---|---|
| | | IDS↑ | FaceNet↑ | IDS(REF)↑ | LPIPS↓ | MUSIQ↑ | NIQE↓ | FID↓ | IDS↑ | FaceNet↑ | IDS(REF)↑ | LPIPS↓ | MUSIQ↑ | NIQE↓ | FID↓ |
| CodeFormer (NeurIPS'22) | 0 | 0.783 | 0.822 | 0.545 | **0.1839** | 75.88 | 4.38 | 31.7 | 0.370 | 0.677 | 0.265 | **0.3113** | 76.12 | 4.30 | 49.6 |
| DiffBIR (ECCV'24) | 0 | 0.831 | 0.842 | 0.575 | 0.2268 | **76.64** | 5.72 | 34.9 | 0.356 | 0.672 | 0.253 | 0.3606 | **75.71** | 6.24 | 55.3 |
| RefLDM (NeurIPS'24) | 1 | 0.826 | 0.837 | 0.624 | 0.2211 | 72.30 | 4.61 | 28.0 | 0.571 | 0.733 | 0.554 | 0.3366 | 74.32 | 4.52 | **36.0** |
| RestorerID (arXiv) | 1 | 0.804 | 0.832 | 0.591 | 0.2350 | 73.35 | 4.98 | 31.0 | 0.411 | 0.690 | 0.408 | 0.4130 | 74.49 | 4.71 | 52.7 |
| Ours | 1 | **0.843** | **0.850** | **0.732** | 0.2054 | 75.29 | **3.96** | **25.5** | **0.609** | **0.743** | **0.712** | 0.3647 | 75.22 | **3.84** | 38.3 |

**Implementation Details.** We employ an LDM (Rombach et al., 2022) backbone with 865M parameters pre-trained on the WebLI (Chen et al., 2023) dataset for text-to-image synthesis. We fine-tuned the VAE following (Hsiao et al., 2024), using the 68411 remaining FFHQ (Karras et al., 2019) images after excluding the FFHQ-Ref (Hsiao et al., 2024) validation and test images. Our model is trained on the FFHQ-Ref training set for 100K steps, with batch size 256 and learning rate `8e-5`. The cross-attention dimension is 1024. To enable classifier-free guidance (Ho & Salimans, 2022; Brooks et al., 2023), we randomly drop the LQ condition as well as the components in Composite Context independently with a 0.1 probability. The Composite Context components are dropped through attention masking. The classifier-guidance scales are selected as $s_i = 1.2$ and $s_c = 1.2$ for inference. The Hard Example Identity Loss balancing parameter $w_{\text{HID}}$ is 0.1 following (Hsiao et al., 2024), and $\lambda$ is set as 0.6 by default. The AdaIN (Karras et al., 2019)-based color fix (Wang et al., 2024b) is applied on the model output as a post-processing step.

**Evaluation.** Following the previous works (Hsiao et al., 2024; Ying et al., 2024; Zhang et al., 2025), we use LPIPS (Zhang et al., 2018) for perceptual similarity, and IDS (*i.e.*, the cosine similarity of ArcFace (Deng et al., 2022) embedding) for person identity preservation. This "IDS" is calculated between the restoration result and the HQ image. Since we optimize the identity loss using the ArcFace (Deng et al., 2022) model during training, using IDS alone may not properly reflect generalization performance due to potential overfitting. Thus, we also evaluate the ArcFace IDS with respect to the first reference face for each LQ test image (denoted as "IDS(REF)"), as well as the FaceNet IDS with respect to HQ (denoted as "FaceNet"). We also use no-reference metrics including MUSIQ (Ke et al., 2021), NIQE (Mittal et al., 2012), and FID (Heusel et al., 2017) for image quality. It is worth noting that, while in the image restoration literature, image quality metrics are important, when it comes to identity-preserving face restoration applications, identity preservation is often prioritized over the image quality For example, consider the scenario of restoring a picture of a memorable moment (e.g. graduation, wedding etc.) – the restoration result is no longer meaningful if the person's identity changed.

## 4.1 Experimental Results and Comparison with SOTA

To validate the effectiveness of our proposed method, we evaluate our method on the FFHQ-Ref test datasets with Moderate and Severe degradations, and CelebA-Ref-Test following (Hsiao et al., 2024). We compare our method with some state-of-the-art no-reference face restoration methods, namely CodeFormer (Zhou et al., 2022) and DiffBIR (Lin et al., 2024), as well as the latest reference-based face restoration methods, namely RefLDM (Hsiao et al., 2024) and RestorerID (Ying et al., 2024). The quantitative results on FFHQ-Ref test datasets can be found in Table 1. The multi-reference results are in Table 2. The quantitative results on CelebA-Ref-Test can be found in Table 3. The visualization for FFHQ-Ref test sets can be found in Figure 3 and Figure 4. All results of the related works are reproduced using their official code and checkpoints.

As shown in Table 1, the IDS and FaceNet are computed between the output and HQ ground-truth, whereas IDS(REF) is computed between the output and the first reference face. The overall trend is that no-reference methods like CodeFormer (Zhou et al., 2022) and DiffBIR (Lin et al., 2024) tend to achieve good perceptual similarity (LPIPS) and image quality (MUSIQ), but worse identity preservation compared to reference-based methods like RefLDM (Hsiao et al., 2024) and RestorerID (Ying et al., 2024). And notably, our model

Table 2: Multi-reference face inference results. The identity preservation improves when the number of reference faces increases Hsiao et al. (2024). Note, the IDS(REF) is calculated using the first reference face, and it may drop with more than one reference face, because the additional reference faces can pull the model output slightly further from the first reference face in Eq. (6). More details including standard deviation are available in Appendix A.3.1.

| #REF | FFHQ-Ref Moderate | | | | | | | FFHQ-Ref Severe | | | | | | |
|---|---|---|---|---|---|---|---|---|---|---|---|---|---|---|
| | IDS↑ | FaceNet↑ | IDS(REF)↑ | LPIPS↓ | MUSIQ↑ | NIQE↓ | FID↓ | IDS↑ | FaceNet↑ | IDS(REF)↑ | LPIPS↓ | MUSIQ↑ | NIQE↓ | FID↓ |
| 1 | 0.843 | 0.850 | **0.732** | 0.2054 | **75.29** | 3.96 | 25.5 | 0.609 | 0.743 | **0.712** | 0.3647 | **75.22** | 3.84 | 38.3 |
| 2 | 0.857 | 0.856 | 0.693 | 0.2042 | 75.28 | **3.95** | 25.4 | 0.640 | 0.752 | 0.650 | 0.3625 | 75.20 | **3.82** | **38.2** |
| 3 | 0.861 | **0.859** | 0.683 | 0.2040 | 75.29 | 3.96 | 25.4 | 0.652 | 0.755 | 0.636 | 0.3619 | 75.19 | **3.82** | 38.4 |
| 4 | **0.863** | **0.859** | 0.680 | 0.2039 | 75.29 | 3.96 | 25.5 | 0.657 | **0.757** | 0.630 | 0.3617 | 75.20 | 3.82 | 38.3 |
| 5 | **0.863** | **0.859** | 0.678 | **0.2038** | 75.29 | 3.96 | 25.5 | **0.658** | **0.757** | 0.626 | **0.3615** | 75.20 | 3.83 | **38.2** |

Table 4: Ablation study on the Composite Context (CC) and Hard Example Identity Loss (HID).

| Modules | | FFHQ-Ref Moderate | | | | | | | FFHQ-Ref Severe | | | | | | |
|---|---|---|---|---|---|---|---|---|---|---|---|---|---|---|---|
| CC | HID | IDS↑ | FaceNet↑ | IDS(REF)↑ | LPIPS↓ | MUSIQ↑ | NIQE↓ | FID↓ | IDS↑ | FaceNet↑ | IDS(REF)↑ | LPIPS↓ | MUSIQ↑ | NIQE↓ | FID↓ |
| - | - | 0.811 | 0.841 | 0.565 | 0.2104 | **76.02** | **3.85** | 26.0 | 0.231 | 0.637 | 0.168 | 0.3896 | 73.85 | **3.67** | 43.4 |
| ✓ | - | 0.822 | 0.847 | 0.584 | 0.2074 | 75.66 | 3.89 | 25.9 | 0.345 | 0.675 | 0.288 | 0.3694 | **75.46** | 3.83 | **38.0** |
| ✓ | ✓ | **0.843** | **0.850** | **0.732** | **0.2054** | 75.29 | 3.96 | **25.5** | **0.609** | **0.743** | **0.712** | 0.3647 | 75.22 | 3.84 | 38.3 |

consistently achieves the best identity preservation (which is the top-priority in the reference-based face restoration task) across all test datasets, while still achieving competitive image quality.

As shown in Table 2, the identity preservation will improve as we introduce more reference faces. The effect saturates at roughly five images, which is similar to the observation in (Hsiao et al., 2024). Note, the IDS(REF) is calculated using the first available reference face. That means the additional reference faces could pull the model output slightly further from the first reference through Eq. (6). Thus, IDS(REF) may drop with additional reference faces. Nevertheless, our worst IDS(REF) is still higher than previous methods in Table 1.

As shown in Figure 3 for FFHQ-Ref Moderate, when the input LQ image contains a moderate degradation, the IDS performance gap among the models is not very large in Table 1, hence it is highly recommended to zoom-in to visually distinguish the differences in restored face details. For instance, the black moles are well preserved on the sixth row in Figure 3. While other methods tends to excessively smooth the skin texture, our model generates more realistic textures.

As shown in Figure 4 for FFHQ-Ref Severe, when the LQ face is almost unrecognizable, our method can still sufficiently leverage the reference face and generate a face that is very close to the ground truth, preserving identity. In contrast, almost every other method generates a visually different person in most cases, which justifies the consistent improvements on the identity metrics of our method.

Table 3: Comparison with previous reference-based methods on CelebA-Ref-Test (Hsiao et al., 2024).

| Method | #REF | CelebA-Ref-Test | | | | | |
|---|---|---|---|---|---|---|---|
| | | IDS↑ | FaceNet↑ | IDS(REF)↑ | LPIPS↓ | MUSIQ↑ | NIQE↓ |
| RefLDM | 1 | 0.768 | 0.821 | 0.564 | 0.2453 | 72.11 | 4.75 |
| RestorerID | 1 | 0.756 | 0.820 | 0.527 | 0.2690 | 74.86 | 5.22 |
| Ours | 1 | **0.779** | **0.827** | **0.691** | **0.2310** | **75.64** | **3.98** |

As demonstrated in Table 3 for CelebA-Ref-Test, our model still achieves the best identity preservation compared to other reference-based methods. All the above experimental results demonstrate the effectiveness of our method, especially in terms of identity preservation.

Table 5: Ablation study on individual components of Composite Context. The evaluation of different combinations is carried out by using different attention masks with the same model checkpoint.

| Composite Context | | FFHQ-Ref Moderate | | | | | | | FFHQ-Ref Severe | | | | | | |
|---|---|---|---|---|---|---|---|---|---|---|---|---|---|---|---|
| High-Level | General | IDS↑ | FaceNet↑ | IDS(REF)↑ | LPIPS↓ | MUSIQ↑ | NIQE↓ | FID↓ | IDS↑ | FaceNet↑ | IDS(REF)↑ | LPIPS↓ | MUSIQ↑ | NIQE↓ | FID↓ |
| - | - | 0.738 | 0.805 | 0.516 | 0.2196 | 74.43 | 3.99 | 27.9 | 0.186 | 0.616 | 0.137 | 0.3875 | 73.03 | 3.92 | 47.4 |
| - | ✓ | 0.770 | 0.821 | 0.567 | 0.2087 | 75.01 | **3.94** | 25.7 | 0.348 | 0.666 | 0.320 | 0.3713 | 74.51 | **3.83** | 40.1 |
| ✓ | - | 0.835 | 0.846 | 0.707 | 0.2094 | 75.20 | 3.98 | 25.9 | 0.535 | 0.717 | 0.625 | 0.3800 | 74.95 | 3.85 | 40.2 |
| ✓ | ✓ | **0.843** | **0.850** | **0.732** | **0.2054** | **75.29** | 3.96 | **25.5** | **0.609** | **0.743** | **0.712** | **0.3647** | **75.22** | 3.84 | **38.3** |

Table 6: Ablation Study on Individual Components of the Hard Identity Loss.

| $\lambda$ | FFHQ-Ref Moderate | | | | | | | FFHQ-Ref Severe | | | | | | |
|---|---|---|---|---|---|---|---|---|---|---|---|---|---|---|
| | IDS↑ | FaceNet↑ | IDS(REF)↑ | LPIPS↓ | MUSIQ↑ | NIQE↓ | FID↓ | IDS↑ | FaceNet↑ | IDS(REF)↑ | LPIPS↓ | MUSIQ↑ | NIQE↓ | FID↓ |
| 0 | **0.844** | **0.855** | 0.621 | 0.2039 | 75.36 | 3.97 | 25.5 | 0.485 | 0.712 | 0.465 | 0.3664 | 75.22 | 3.85 | 38.5 |
| 0.6 | 0.843 | 0.850 | 0.732 | 0.2054 | 75.29 | 3.96 | 25.5 | **0.609** | **0.743** | 0.712 | 0.3647 | 75.22 | 3.84 | 38.3 |
| 1 | 0.779 | 0.821 | **0.794** | 0.2076 | 75.43 | 3.95 | 25.6 | 0.605 | 0.742 | **0.768** | 0.3666 | 75.29 | 3.90 | 38.8 |

## 4.2 Ablation Study and Discussions

We conduct the ablation study in a hierarchical way, firstly, coarse-grained based on the two Composite Context and Hard Example Identity Loss modules. Then we conduct the fine-grained ablation study for each component in these modules.

**Module-wise Ablation.** Since the two modules are independent of each other, we conduct the ablation study by removing some of them, and then retrain the model. As shown in Table 4, both the context and loss contribute significantly to the final performance, because the removal of any of them will lead to a major performance drop. Removing both makes the model degenerate into a no-reference face restoration model, which lags behind our model too much in identity preservation. This means both Composite Context and Hard Example Identity Loss are effective. Next, we conduct an ablation study on the individual components of these modules.

**Composite Context Ablation.** As shown in Table 5, we study the contribution of individual components in the Composite Context by applying attention masks during inference. It can be seen in the table that all the multi-level components, including high-level and general components clearly contribute significantly to the final performance, as the removal of any of them will lead to a performance drop, especially on the FFHQ-Ref Severe test dataset.

**Hard Example Identity Loss Ablation.** As shown in Table 6, we conduct an ablation study on the individual components in the Hard Example Identity Loss, by adjusting the balancing parameter $\lambda$ between the HQ image and the reference image in Eq. (3). According to the results, when we only use the ID loss with the HQ image ($\lambda = 0$), the IDS(REF) is much lower, so is IDS on FFHQ-Ref Severe. When we only use the ID loss with the REF image ($\lambda = 1$), the IDS will be traded off with IDS(REF). Hence, we empirically set the $\lambda$ parameter as 0.6 by default, by considering all the three identity preservation metrics. The case where the Hard Identity Loss is removed ($w_{HID} = 0$) is at the second row of Table 4, and that leads to a much lower performance regardless of the $\lambda$ parameter.

**Influence of Reference Face.** The above ablation study supports the effectiveness of our method when using a *correct* reference face. While the problem of reference-based face restoration assumes a reference face with correct identity is provided, it is difficult to guarantee in real-world applications. To demonstrate the influence of the reference face, we deliberately use a *wrong* reference face, as shown in Figure 5. According to our observation, when the input LQ image has moderate information loss with the person identity roughly recognizable, our model will largely follow the LQ, and add slight identity-related details to the result, as shown in the first row in the figure. When the input LQ has severe information loss with the person identity almost unrecognizable, the REF face image becomes dominant and show stronger impact in the resulting image.

This phenomenon, on the one hand, further demonstrates the effectiveness of our method through the influence of the reference face. On the other hand, it also implies the importance of ensuring the correct identity in real-world applications for reference-based face restoration.

**Limitations and Future Work. (1)** The training data is from simulated degradation pipelines (Wang et al., 2021b), which means the model may underperform on in-the-wild face images with unknown degradations. **(2)** While this reference-based task assume high-quality reference images are available, in practical scenarios the reference image quality may vary. Figuring out which reference face among an album is most helpful could be a direction for future exploration. **(3)** A large-scale high-quality dataset for this reference-based task is still missing, and the FFHQ-Ref (Hsiao et al., 2024) training set only contains 18816 images. Potential approaches for more data could be filtering face recognition datasets (Zhu et al., 2021) or video frames. We leave these directions for future study.

| LQ | REF | Result | HQ |
|---|---|---|---|

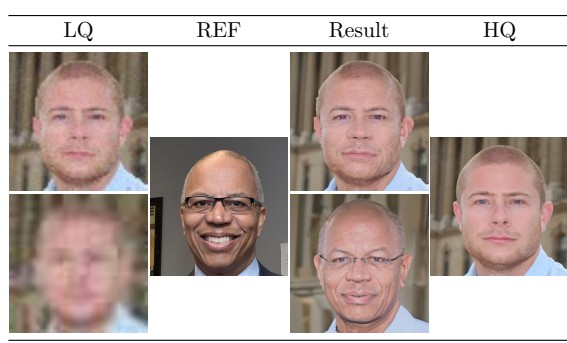

Figure 5: Demonstration of the impact of reference face image, by deliberately supplying the model with a reference face of a wrong identity. The first row is from FFHQ-Ref Moderate, and the second row is from FFHQ-Ref Severe.

## 5  Conclusion

We present a reference-based face restoration method, highlighting two key modules: Composite Context and Hard Example Identity Loss that focus on identity preservation. The two key modules are designed to better exploit reference face images, while all the existing works leverage it to a lesser extent. Meanwhile, the proposed method can be extended for the multi-reference case in a training-free manner. Experimental results on the FFHQ-Ref and CelebA-Ref-Test datasets demonstrate the effectiveness of our proposed method. Ablation studies on the Composite Context and Hard Example Identity Loss suggest that all the proposed modules in our method, including the individual components in the modules, are effective and make a considerable impact on identity preservation.

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

## Contents

# A    Appendix: Additional Experimental Results and Discussions

## A.1    LLM Usage, Ethics Statement and Broader Impact

LLM is used to fix typo as well as grammar error, and polish language for this manuscript.

Our method aims at restoring low-quality face images, with the goal of contributing positively to society. However, being a diffusion-based face generative model, it might be abused to forge DeepFake (Pei et al., 2024) images. We suggest real-world face restoration service providers apply invisible watermarks (Fernandez et al., 2024) to the generated result to mitigate potential risks and negative societal impact.

## A.2    Key Differences Compared to The Most Relevant Works

In this paper, we present a simple yet effective reference-based face image restoration method. While seemingly straightforward, the proposed method is not merely an adaptation of previous works; it is underpinned by strong motivations. The insights provided by this method are novel and have the potential to inspire future research in this field.

This paper focuses on reference-based face restoration. The most related works to this paper are RefLDM (Hsiao et al., 2024), RestorerID (Ying et al., 2024), and InstantRestore (Zhang et al., 2025). The key differences between our method and the previous methods are:

- **Ensemble representation instead of partial information through face feature.** The previous methods only use a single representation for the reference face, which only covers partial information of the reference and does not maximize the utilization of the reference. In contrast, our Composite Context combines multi-level face-specific representations to comprehensively exploit the information in the reference face. The Composite Context conceptually resembles (Mei et al., 2025; Wang et al., 2024a) which employ multiple modalities to aid image restoration, and (Podell et al., 2024) which concatenates two text representations for text-to-image synthesis. Our ablation studies suggest significant contribution from every single component in the proposed composite context.

- **Addressing the overlooked learning inefficiency issue in identity loss.** While many related works (Hsiao et al., 2024; Wang et al., 2025; Zhang et al., 2025) incorporate the identity loss, a notable learning inefficiency issue where the loss value plateaus at a tiny value (indicates learning inefficiency in the context of metric learning (Musgrave et al., 2020; Roth et al., 2020; Schroff et al., 2015)) has been overlooked. Hard example mining is a known technique in metric learning. However, we uniquely apply it to enhance identity preservation in reference-based face restoration, offering a novel and targeted improvement over the standard identity loss functions typically employed. This resolves a long-overlooked learning inefficiency issue. The ablation studies in the paper show a significant performance gain compared to the original version of identity loss.

- **Alleviating reliance on multi-ref training data.** Some existing methods support only one reference face (Ying et al., 2024). Some others support multiple (Hsiao et al., 2024), but also require multiple reference face images during training. It is noted that requiring more than one reference face makes training data collection difficult. In contrast, even if our method only uses one reference face during training, our model can be adapted to effectively support multiple reference faces during inference in a training-free manner. Thus, we demonstrate that the paradigm of "training with a single reference, inference with multiple references" offers a more scalable approach, considering the practical challenges in dataset collection and model training for reference-based face restoration.

## A.3    Detailed Experimental Results of Our Method

### A.3.1    Detailed Quantitative Results and More Visualizations

The detailed results and comparison with state-of-the-art methods on FFHQ-Ref Moderate, FFHQ-Ref Severe, and CelebA-Ref-Test can be found in Table 7, Table 8, and Table 9, respectively. Additional visualizations are provided in Figure 8, Figure 9, and Figure 10 for those three test sets.

The detailed multi-reference face restoration results and comparison with state-of-the-art methods can be found in Table 10, Table 11, and Table 12, respectively for the three test sets. Some visualizations are provided in Figure 13 to demonstrate the effect of using more than one reference faces.

### A.3.2 Qualitative Ablation of Each Module

According to our visualizations on the FFHQ-Ref Severe test dataset, each component also makes a qualitative difference beyond the quantitative improvements presented in the paper. We observed that:

- **Composite Context:** As indicated in our quantitative ablation studies, further removing the composite context introduces even more artifacts, along with distortions to facial parts and shape in the resulting image. We also observe skin texture artifacts, incorrect eye colors, and occasionally, an incomplete facial structure. Overall, this significantly reduces the resemblance to the HQ/REF face. These visual degradations are consistent with the quantitative results. Some examples of qualitative ablation can be found in Figure 11.

- **Hard Example Identity Loss:** According to our ablation studies, removing this loss sometimes introduces artifacts around the eyes and lips in the resulting face image, or causes face shape distortions and uneven eye sizes. Overall, the resulting face appears less similar to the HQ/REF image. Given that ArcFace embeddings are sensitive to these key facial features, these visual observations are consistent with the quantitative results. Some examples of qualitative ablation can be found in Figure 12.

- **Multi-reference Inference:** Visualizations employing multiple reference faces are provided in the previous subsection. Overall, utilizing more reference faces leads to more effective identity preservation.

These observations demonstrate that all components in our proposed method yield qualitative benefits, not just quantitative gains.

### A.3.3 Experiments on Different Classifier-Free Guidance Scale Parameter

A minor ablation study on the classifier-free guidance scale parameter can be found in Table 19.

### A.3.4 Additional Visualizations for Wrong-Reference Ablation

Additional visualizations with wrong reference face (as discussed in Section 4.2) can be found in Figure 14, Figure 15, and Figure 16.

### A.3.5 Robustness Against Pose Difference Between LQ and REF

In practice, the face pose, angle, lighting, and expression are open challenges in reference-based face restoration, as also mentioned by InstantRestore (Zhang et al., 2025) in their Figure 9 on limitations. Nevertheless, existing reference-based face restoration benchmark datasets are not specifically designed to reflect challenges such as large pose variations, since most faces in these datasets are near-frontal. In this paper, we adopt the same problem setting, datasets, and evaluation protocol as RefLDM (NeurIPS 2024).

To validate our method's robustness against pose differences between the LQ and REF images, we grouped the FFHQ-Ref Severe test dataset into several ranges based on the absolute difference in yaw angle between the original high-quality (HQ) and reference (REF) images. This difference, measured in degrees, is calculated as "abs(yaw(HQ)-yaw(REF))". We then re-calculated the quantitative metrics for each group.

From the above results in Table 13, our proposed method demonstrates greater robustness than the previous state-of-the-art, RefLDM (NeurIPS 2024), against the face pose challenge.

### A.3.6 Robustness Against REF Image Quality Changes

To evaluate our method's robustness to degraded reference images, we used the FFHQ-Ref Severe test dataset. For this dataset, reference images were deterministically degraded using Gaussian blur with fixed kernel sizes (for a better controlled experiment). The results, including a comparison with RefLDM, are presented in Table 14.

In the table, "IDS" and "FaceNet" denote the ArcFace and FaceNet cosine similarities, respectively, between the high-quality (HQ) image and the restoration result. "IDS(REF)" represents the ArcFace cosine similarity between the first reference face and the result. The results indicate that although performance naturally degrades with lower-quality reference images, our model exhibits much greater robustness compared to the prior state-of-the-art method, RefLDM.

### A.3.7 Evaluation on Real-World LQ Face Images

In order to evaluate our model's performance on real-world low-quality face images, namely face images with non-synthetic degradation, we collected a small test set of 65 images from 21 different individuals using a Google Pixel mobile phone. The low-quality (LQ) images in this set exhibit real-world degradations, including blur and noise resulting from motion, defocus, or low-light conditions. Notably, the corresponding reference images are also not of perfect quality, and ground truth high-quality (HQ) images are unavailable for this set. The quantitative experimental results are presented in Table 15 (note: some metrics cannot be computed due to the absence of ground truth). These results, in fact, show that our method is state-of-the-art, even for very challenging real-world degradations.

Since the volunteers for photo donation involve the authors of this manuscript, we are unable to show the visualization for keeping this manuscript anonymous.

### A.3.8 Human Subject Evaluation

While our method already significantly outperforms the state-of-the-art RefLDM on synthetic benchmarks and real-world data (the 65 test images in the previous subsection), a subjective human evaluation would further strengthen the effectiveness of our method. To that end, we performed two blind (with model names hidden), side-by-side user studies between RefLDM and our method.

First, six participants evaluated our method against RefLDM on 65 real-world test images (with real Pixel phone camera degradations; see the previous subsection). Based on identity preservation, our method was preferred in 63.5% of evaluations (248/390), RefLDM was preferred in 11.0% (43/390), and the results were a tie in 25.3% (99/390).

Second, the same six participants evaluated 50 random samples each from the FFHQ-Ref Severe dataset. In this test, our method was rated higher for identity preservation in 47.6% of cases (143/300), while RefLDM was rated higher in 29.0% (87/300), with 23.3% of results being a tie (70/300).

These human evaluations confirm that our method outperforms RefLDM in preserving identity across both real-world and synthetically degraded images.

### A.3.9 Inference Time Cost

Our diffusion model backbone is an 865M-parameter LDM, almost identical to Stable Diffusion v1.5. On an Nvidia A5000 GPU, the inference time per image is 7.18 seconds, with CUDA memory usage at 8.7GB. The feature extraction components contribute minimally to this total time: ViT-B/16 (FaRL) takes 0.008 seconds, and ArcFace (ResNet-50) takes 0.010 seconds.

### A.4   Comparison with Additional Related Works

### A.4.1   Comparison with InstantRestore (reference-based)

InstantRestore (Zhang et al., 2025) utilizes the CelebRef-HQ dataset for training and evaluation. In our work, we employ a similar benchmark, CelebA-Ref-Test, which was curated by RefLDM (NeurIPS 2024) from the CelebA-HQ dataset.

The comparison between InstantRestore and our method is available in Table 6. According to the results, while InstantRestore indeed achieve a slightly higher IDS as Arcface is used in its training procedure, its FaceNet metric and IDS(REF) metric are surpassed by our method, especially by a large margin in IDS(REF). In particular, as both ground-truth image (used by IDS) and the reference image (used by IDS(REF)) are the same person, a better method should achieve high performance in both IDS and IDS(REF). That suggests that InstantRestore generates faces that look like the same person as the ground-truth face, but not necessarily the same person as the reference face (even though they are the same person). In contrast, our method achieves higher IDS(REF) than InstantRestore by a large margin. Besides, our method outperforms InstantRestore in image quality, as almost all quality metrics of our method are better than those of InstantRestore on FFHQ-Ref Moderate and FFHQ-Ref Severe.

InstantRestore also created an additional non-celebrity test set, although it remains unpublished at the time of writing. However, we were able to extract some low-quality/reference (LQ/REF) pairs from InstantRestore's "additional test images" by examining their publicly available arXiv LaTeX source code (specifically, from the `images/common_people_results` directory). This process yielded a subset of 9 test images, each associated with two reference faces. The quantitative results for this subset are provided in Table 16. And the visualization of all images are available in Figure 7. Note that some metrics are unavailable due to the absence of ground truth HQ images. Despite the small size of this subset, our method clearly outperforms InstantRestore.

### A.4.2   Comparison with OSDFace (no-reference)

OSDFace Wang et al. (2025) is a no-reference face restoration method. The comparison of OSDFace and our method is avalable in Table 17. Compard with our method, OSDFace can indeed achieve good image quality as reflected by most image quality metrics. However, its identity preservation falls behind our method especially on the FFHQ-Ref Severe dataset.

### A.4.3   Comparison with InterLCM (no-reference)

It is important to note that InterLCM (Li et al., 2025) is a no-reference method; it does not use any reference face images for restoration and thus operates under a different problem setting than our reference-based approach. Despite this fundamental difference, we compared our method with InterLCM on the FFHQ-Ref Severe test dataset to provide a performance benchmark. The results are presented in Table 18. The degradations in the FFHQ-Ref Severe dataset proved too challenging for the officially pre-trained InterLCM model. Regarding computational cost, using an Nvidia A5000 GPU, InterLCM takes 0.106 seconds per image for inference and consumes 9.7GB of CUDA memory. The computational cost of our proposed method is detailed in our response to the previous question. We will incorporate these InterLCM cost details into the manuscript in the next revision.

Other related restoration works include VQFR, DAEFR, and DMDNet (Li et al., 2022). Since these methods were already compared and significantly outperformed in the RefLDM paper, we omitted them from our comparisons for brevity.

### A.5   Discussion: Design Details

### A.5.1   Architecture and Implementation Details

Our model architecture closely follows Stable Diffusion v1.5, an 865M-parameter Latent Diffusion Model (LDM), with two modifications: (1) the VAE latent size is $(64, 64, 8)$, as indicated in the overview diagram,

and (2) the UNet's cross-attention dimension is 1024. Training details (including dataset, batch size, learning rate, loss weights, and total training iterations) are provided in the "Implementation Details" part of Section 4. This section also covers key inference details, such as classifier-free guidance parameters. Additionally, we use DDIM with 50 steps for sampling.

### A.5.2 Scarcity of Multi-Reference Training Data

Supporting multiple reference faces fundamentally through network architecture presents data scalability issues. Since collecting a high-quality reference-based face restoration dataset is challenging, almost half of the samples in the FFHQ-Ref dataset have only a single reference face. In particular, in the FFHQ-Ref training dataset, 8351 out of 18816 samples (44.3%) have only a single reference face; 3670 samples (19.5%) have two; and 1749 samples (9.3%) have three. This means roughly 73.2% of the training data have only three or fewer reference faces. Given this challenge in dataset collection, we propose that "training with a single reference face, while supporting multiple reference faces during inference" is a more scalable design.

### A.5.3 Identity Loss's Influence to Image Quality

The identity loss has a minor impact on no-reference image quality metrics, an issue also noted in Section 3.2 of RefLDM (Hsiao et al., 2024). We therefore use 0.1 as the hard example identity loss balancing parameter $w_{\text{HID}}$ following RefLDM's choice for their original identity loss. Furthermore, the influence of the identity loss on image quality is less than the standard deviation of the image quality metrics themselves, as detailed in Tables 7 and 8 in this supplementary material.

In some real-world applications, identity preservation can be more important than perceptual quality. For instance, when restoring the face in a user's selfie, it would be worse if the restoration model turned the person into someone else. In such cases, preserving identity at a subtle cost to image quality is a worthy trade-off.

### A.5.4 Attempt on Other Face Representations

Face representation is a crucial component, and indeed, existing methods in this area are quite mature. During our explorations, we experimented with using local patches cropped around facial landmarks from the reference image, processed by small neural networks, to form a low-level representation. However, our experimental results indicated that the FaRL representation is sufficient on its own, likely because it also effectively encodes low-level information, rendering the explicit patch-based features redundant.

Table 7: Detailed Quantitative Results on FFHQ-Ref Moderate test set.

| Method | #REF | FFHQ-Ref Moderate | | | | | | |
|---|---|---|---|---|---|---|---|---|
| | | IDS | FaceNet | IDS(REF) | LPIPS | MUSIQ | NIQE | FID |
| CodeFormer | 0 | $0.783 \pm 0.082$ | $0.822 \pm 0.047$ | $0.545 \pm 0.106$ | $0.1839 \pm 0.0471$ | $75.88 \pm 2.01$ | $4.38 \pm 0.69$ | 31.7 |
| DiffBIR | 0 | $0.831 \pm 0.095$ | $0.842 \pm 0.056$ | $0.575 \pm 0.108$ | $0.2268 \pm 0.0633$ | $76.64 \pm 1.64$ | $5.72 \pm 1.23$ | 34.9 |
| RefLDM | 1 | $0.826 \pm 0.077$ | $0.837 \pm 0.048$ | $0.624 \pm 0.096$ | $0.2210 \pm 0.0583$ | $72.30 \pm 4.89$ | $4.61 \pm 0.64$ | 28.0 |
| RestorerID | 1 | $0.804 \pm 0.099$ | $0.832 \pm 0.054$ | $0.591 \pm 0.096$ | $0.2350 \pm 0.0688$ | $73.35 \pm 5.12$ | $4.98 \pm 0.81$ | 31.0 |
| Ours | 1 | $0.843 \pm 0.076$ | $0.850 \pm 0.051$ | $0.732 \pm 0.069$ | $0.2054 \pm 0.0606$ | $75.29 \pm 2.77$ | $3.96 \pm 0.71$ | 25.5 |

Table 8: Detailed Quantitative Results on FFHQ-Ref Severe test set.

| Method | #REF | FFHQ-Ref Severe | | | | | | |
|---|---|---|---|---|---|---|---|---|
| | | IDS | FaceNet | IDS(REF) | LPIPS | MUSIQ | NIQE | FID |
| CodeFormer | 0 | $0.370 \pm 0.150$ | $0.677 \pm 0.061$ | $0.265 \pm 0.132$ | $0.3113 \pm 0.0801$ | $76.12 \pm 1.94$ | $4.30 \pm 0.70$ | 49.6 |
| DiffBIR | 0 | $0.356 \pm 0.144$ | $0.672 \pm 0.058$ | $0.253 \pm 0.124$ | $0.3606 \pm 0.0879$ | $75.71 \pm 2.81$ | $6.24 \pm 1.22$ | 55.3 |
| RefLDM | 1 | $0.571 \pm 0.110$ | $0.733 \pm 0.052$ | $0.554 \pm 0.112$ | $0.3366 \pm 0.0756$ | $74.32 \pm 3.36$ | $4.52 \pm 0.62$ | 36.0 |
| RestorerID | 1 | $0.411 \pm 0.110$ | $0.690 \pm 0.052$ | $0.408 \pm 0.103$ | $0.4130 \pm 0.0741$ | $74.49 \pm 3.41$ | $4.71 \pm 0.65$ | 52.7 |
| Ours | 1 | $0.609 \pm 0.089$ | $0.743 \pm 0.048$ | $0.712 \pm 0.068$ | $0.3647 \pm 0.0722$ | $75.22 \pm 2.46$ | $3.84 \pm 0.64$ | 38.3 |

Table 9: Detailed Quantitative Results on CelebA-Ref-Test test set.

| Method | #REF | CelebA-Ref-Test | | | | | | |
|---|---|---|---|---|---|---|---|---|
| | | IDS | FaceNet | IDS(REF) | LPIPS | MUSIQ | NIQE | FID |
| RefLDM | 1 | $0.768 \pm 0.085$ | $0.821 \pm 0.046$ | $0.564 \pm 0.096$ | $0.2453 \pm 0.0550$ | $72.11 \pm 4.59$ | $4.75 \pm 0.55$ | 19.4 |
| RestorerID | 1 | $0.756 \pm 0.098$ | $0.820 \pm 0.049$ | $0.527 \pm 0.090$ | $0.2690 \pm 0.0629$ | $74.86 \pm 3.82$ | $5.22 \pm 0.76$ | 25.4 |
| Ours | 1 | $0.779 \pm 0.086$ | $0.827 \pm 0.048$ | $0.691 \pm 0.064$ | $0.2310 \pm 0.0540$ | $75.64 \pm 2.44$ | $3.98 \pm 0.53$ | 18.4 |

Table 10: Detailed Quantitative Results on FFHQ-Ref Moderate test set. Note, our multi-reference face support is training-free, while RefLDM's is not.

| Method | #REF | FFHQ-Ref Moderate | | | | | | |
|---|---|---|---|---|---|---|---|---|
| | | IDS | FaceNet | IDS(REF) | LPIPS | MUSIQ | NIQE | FID |
| RefLDM | 1 | $0.826 \pm 0.077$ | $0.837 \pm 0.048$ | $0.624 \pm 0.096$ | $0.2210 \pm 0.0583$ | $72.30 \pm 4.89$ | $4.61 \pm 0.64$ | 28.0 |
| Ours | 1 | $0.843 \pm 0.076$ | $0.850 \pm 0.051$ | $0.732 \pm 0.069$ | $0.2054 \pm 0.0606$ | $75.29 \pm 2.77$ | $3.96 \pm 0.71$ | 25.5 |
| RefLDM | 2 | $0.839 \pm 0.067$ | $0.844 \pm 0.045$ | $0.630 \pm 0.094$ | $0.2150 \pm 0.0577$ | $73.25 \pm 4.34$ | $4.57 \pm 0.62$ | 27.6 |
| Ours | 2 | $0.857 \pm 0.069$ | $0.856 \pm 0.049$ | $0.693 \pm 0.075$ | $0.2042 \pm 0.0603$ | $75.28 \pm 2.75$ | $3.95 \pm 0.71$ | 25.4 |
| RefLDM | 3 | $0.845 \pm 0.063$ | $0.847 \pm 0.045$ | $0.635 \pm 0.092$ | $0.2117 \pm 0.0574$ | $73.87 \pm 3.92$ | $4.53 \pm 0.63$ | 27.2 |
| Ours | 3 | $0.861 \pm 0.067$ | $0.859 \pm 0.049$ | $0.683 \pm 0.077$ | $0.2040 \pm 0.0602$ | $75.29 \pm 2.75$ | $3.96 \pm 0.71$ | 25.5 |
| RefLDM | 4 | $0.848 \pm 0.061$ | $0.848 \pm 0.044$ | $0.639 \pm 0.090$ | $0.2101 \pm 0.0573$ | $74.26 \pm 3.66$ | $4.50 \pm 0.63$ | 27.2 |
| Ours | 4 | $0.863 \pm 0.066$ | $0.859 \pm 0.049$ | $0.680 \pm 0.078$ | $0.2039 \pm 0.0602$ | $75.29 \pm 2.75$ | $3.96 \pm 0.71$ | 25.5 |
| RefLDM | 5 | $0.848 \pm 0.060$ | $0.848 \pm 0.043$ | $0.641 \pm 0.090$ | $0.2097 \pm 0.0574$ | $74.51 \pm 3.52$ | $4.48 \pm 0.64$ | 27.1 |
| Ours | 5 | $0.863 \pm 0.066$ | $0.859 \pm 0.048$ | $0.678 \pm 0.079$ | $0.2038 \pm 0.0601$ | $75.29 \pm 2.75$ | $3.96 \pm 0.71$ | 25.5 |

Table 11: Detailed Quantitative Results on FFHQ-Ref Severe test set. Note, our multi-reference face support is training-free, while RefLDM's is not.

| Method | #REF | FFHQ-Ref Severe | | | | | | |
|---|---|---|---|---|---|---|---|---|
| | | IDS | FaceNet | IDS(REF) | LPIPS | MUSIQ | NIQE | FID |
| RefLDM | 1 | $0.571 \pm 0.110$ | $0.733 \pm 0.052$ | $0.554 \pm 0.112$ | $0.3366 \pm 0.0756$ | $74.32 \pm 3.36$ | $4.52 \pm 0.62$ | 36.0 |
| Ours | 1 | $0.609 \pm 0.089$ | $0.743 \pm 0.048$ | $0.712 \pm 0.068$ | $0.3647 \pm 0.0722$ | $75.22 \pm 2.46$ | $3.84 \pm 0.64$ | 38.3 |
| RefLDM | 2 | $0.631 \pm 0.091$ | $0.754 \pm 0.049$ | $0.576 \pm 0.100$ | $0.3271 \pm 0.0745$ | $74.82 \pm 3.20$ | $4.51 \pm 0.62$ | 35.4 |
| Ours | 2 | $0.640 \pm 0.078$ | $0.752 \pm 0.047$ | $0.650 \pm 0.073$ | $0.3625 \pm 0.0717$ | $75.20 \pm 2.42$ | $3.82 \pm 0.63$ | 38.2 |
| RefLDM | 3 | $0.662 \pm 0.084$ | $0.764 \pm 0.047$ | $0.594 \pm 0.095$ | $0.3228 \pm 0.0740$ | $75.22 \pm 2.90$ | $4.49 \pm 0.64$ | 35.1 |
| Ours | 3 | $0.652 \pm 0.075$ | $0.755 \pm 0.047$ | $0.636 \pm 0.074$ | $0.3619 \pm 0.0715$ | $75.19 \pm 2.46$ | $3.82 \pm 0.63$ | 38.4 |
| RefLDM | 4 | $0.677 \pm 0.080$ | $0.769 \pm 0.047$ | $0.604 \pm 0.093$ | $0.3203 \pm 0.0731$ | $75.46 \pm 2.73$ | $4.46 \pm 0.64$ | 34.7 |
| Ours | 4 | $0.657 \pm 0.074$ | $0.757 \pm 0.048$ | $0.630 \pm 0.075$ | $0.3617 \pm 0.0715$ | $75.20 \pm 2.42$ | $3.82 \pm 0.63$ | 38.3 |
| RefLDM | 5 | $0.685 \pm 0.078$ | $0.772 \pm 0.048$ | $0.611 \pm 0.091$ | $0.3201 \pm 0.0733$ | $75.62 \pm 2.68$ | $4.46 \pm 0.66$ | 34.7 |
| Ours | 5 | $0.658 \pm 0.074$ | $0.757 \pm 0.049$ | $0.626 \pm 0.077$ | $0.3615 \pm 0.0714$ | $75.20 \pm 2.42$ | $3.83 \pm 0.63$ | 38.2 |

Table 12: Detailed Quantitative Results on CelebA-Ref-Test test set. Note, our multi-reference face support is training-free, while RefLDM's is not.

| Method | #REF | CelebA-Ref-Test | | | | | | |
|--------|------|------|---------|---------|-------|-------|------|-----|
|        |      | IDS | FaceNet | IDS(REF) | LPIPS | MUSIQ | NIQE | FID |
| RefLDM | 1 | $0.768 \pm 0.085$ | $0.821 \pm 0.046$ | $0.564 \pm 0.096$ | $0.2453 \pm 0.0550$ | $72.11 \pm 4.59$ | $4.75 \pm 0.55$ | 19.4 |
| Ours   | 1 | $0.779 \pm 0.086$ | $0.827 \pm 0.048$ | $0.691 \pm 0.064$ | $0.2310 \pm 0.0540$ | $75.64 \pm 2.44$ | $3.98 \pm 0.53$ | 18.4 |
| RefLDM | 2 | $0.775 \pm 0.081$ | $0.824 \pm 0.045$ | $0.580 \pm 0.095$ | $0.2428 \pm 0.0545$ | $73.01 \pm 4.18$ | $4.69 \pm 0.57$ | 18.8 |
| Ours   | 2 | $0.787 \pm 0.084$ | $0.831 \pm 0.047$ | $0.675 \pm 0.071$ | $0.2305 \pm 0.0540$ | $75.65 \pm 2.43$ | $3.98 \pm 0.53$ | 18.4 |
| RefLDM | 3 | $0.774 \pm 0.080$ | $0.824 \pm 0.044$ | $0.587 \pm 0.095$ | $0.2426 \pm 0.0542$ | $73.46 \pm 4.02$ | $4.63 \pm 0.56$ | 18.4 |
| Ours   | 3 | $0.787 \pm 0.084$ | $0.831 \pm 0.047$ | $0.668 \pm 0.076$ | $0.2305 \pm 0.0540$ | $75.65 \pm 2.43$ | $3.98 \pm 0.53$ | 18.4 |
| RefLDM | 4 | $0.771 \pm 0.080$ | $0.824 \pm 0.044$ | $0.591 \pm 0.095$ | $0.2434 \pm 0.0542$ | $73.73 \pm 3.93$ | $4.59 \pm 0.57$ | 18.1 |
| Ours   | 4 | $0.786 \pm 0.084$ | $0.831 \pm 0.047$ | $0.664 \pm 0.080$ | $0.2305 \pm 0.0540$ | $75.65 \pm 2.43$ | $3.98 \pm 0.53$ | 18.4 |
| RefLDM | 5 | $0.767 \pm 0.081$ | $0.822 \pm 0.045$ | $0.594 \pm 0.096$ | $0.2445 \pm 0.0542$ | $73.93 \pm 3.88$ | $4.56 \pm 0.57$ | 18.0 |
| Ours   | 5 | $0.785 \pm 0.085$ | $0.830 \pm 0.046$ | $0.661 \pm 0.082$ | $0.2306 \pm 0.0540$ | $75.65 \pm 2.43$ | $3.98 \pm 0.53$ | 18.4 |

Table 13: Robustness against the face pose (yaw angle) difference between LQ face and REF face. Our method is more robust against the face pose difference than RefLDM.

| Method | Yaw angle diff (deg) | Number of test samples (out of 857) | IDS | FaceNet | IDS(REF) | LPIPS↓ | MUSIQ | NIQE↓ |
|--------|------|------|-----|---------|----------|--------|-------|-------|
| RefLDM | $[0, 15)$ | 530 (61.8%) | 0.584 | 0.736 | 0.571 | **0.3373** | 74.55 | 4.51 |
| Ours   | $[0, 15)$ | 530 (61.8%) | **0.619** | **0.745** | **0.724** | 0.3653 | **75.32** | **3.85** |
| RefLDM | $[15, 30)$ | 231 (26.9%) | 0.562 | 0.731 | 0.540 | **0.3349** | 74.02 | 4.53 |
| Ours   | $[15, 30)$ | 231 (26.9%) | **0.604** | **0.744** | **0.706** | 0.3607 | **74.90** | **3.85** |
| RefLDM | $[30, 90)$ | 96 (11.2%) | 0.512 | 0.717 | 0.493 | **0.3369** | 73.77 | 4.56 |
| Ours   | $[30, 90)$ | 96 (11.2%) | **0.564** | **0.729** | **0.661** | 0.3711 | **75.43** | **3.71** |

Table 14: Robustness against the REF face image quality change.

| Method | Gaussian kernel size | IDS | FaceNet | IDS(REF) | LPIPS↓ | MUSIQ | NIQE↓ | FID↓ |
|--------|------|-----|---------|----------|--------|-------|-------|------|
| RefLDM | 0 | 0.571 | 0.733 | 0.554 | 0.3366 | 74.32 | 4.52 | 36.0 |
| Ours   | 0 | 0.609 | 0.743 | 0.712 | 0.3647 | 75.22 | 3.84 | 38.3 |
| RefLDM | 2 | 0.556 | 0.728 | 0.539 | 0.3505 | 67.84 | 4.82 | 37.5 |
| Ours   | 2 | 0.606 | 0.742 | 0.703 | 0.3682 | 74.84 | 3.95 | 41.0 |
| RefLDM | 4 | 0.509 | 0.712 | 0.485 | 0.3651 | 64.50 | 5.01 | 40.6 |
| Ours   | 4 | 0.587 | 0.735 | 0.677 | 0.3705 | 74.56 | 4.01 | 41.7 |
| RefLDM | 6 | 0.461 | 0.696 | 0.429 | 0.3724 | 64.25 | 5.08 | 42.6 |
| Ours   | 6 | 0.555 | 0.726 | 0.631 | 0.3726 | 74.44 | 4.04 | 42.1 |
| RefLDM | 8 | 0.405 | 0.679 | 0.363 | 0.3759 | 63.52 | 5.09 | 44.3 |
| Ours   | 8 | 0.512 | 0.715 | 0.569 | 0.3744 | 74.34 | 4.05 | 42.8 |

Table 15: Evaluation on Real-World LQ Face Images Captured using Google Pixel Phone.

| Method | IDS(REF) | FaceNet(REF) | MUSIQ | NIQE ↓ |
|--------|----------|--------------|-------|--------|
| RefLDM | 0.447 | 0.741 | 56.10 | 4.21 |
| Ours   | 0.501 | 0.762 | 62.38 | 4.08 |

Table 16: Comparison against InstantRestore on a small set of images.

| Method | IDS(REF) | FaceNet(REF) | MUSIQ | NIQE ↓ |
|---|---|---|---|---|
| InstantRestore | 0.563 | 0.711 | 62.70 | 4.85 |
| Ours | 0.601 | 0.732 | 72.32 | 3.64 |

Figure 6: Quantitative comparison with InstantRestore Zhang et al. (2025) with one reference face image.

| Dataset | Method | IDS↑ | FaceNet↑ | IDS(REF)↑ | LPIPS↓ | MUSIQ↑ | NIQE↓ | FID↓ |
|---|---|---|---|---|---|---|---|---|
| FFHQ-Ref Moderate | InstantRestore | 0.852 | 0.845 | 0.664 | 0.2165 | 69.85 | 6.08 | 34.7 |
| | Ours | 0.843 | 0.850 | 0.732 | 0.2054 | 75.29 | 3.96 | 25.5 |
| FFHQ-Ref Severe | InstantRestore | 0.620 | 0.739 | 0.623 | 0.3296 | 70.09 | 6.35 | 49.1 |
| | Ours | 0.609 | 0.743 | 0.712 | 0.3647 | 75.22 | 3.84 | 38.3 |

Table 17: Quantitative omparison with OSDFace Wang et al. (2025).

| Dataset | Method | IDS↑ | FaceNet↑ | IDS(REF)↑ | LPIPS↓ | MUSIQ↑ | NIQE↓ | FID↓ |
|---|---|---|---|---|---|---|---|---|
| FFHQ-Ref Moderate | OSDFace | 0.830 | 0.839 | 0.577 | 0.2147 | 75.91 | 3.79 | 26.7 |
| | Ours (1-Ref) | 0.843 | 0.850 | 0.732 | 0.2054 | 75.29 | 3.96 | 25.5 |
| FFHQ-Ref Severe | OSDFace | 0.394 | 0.679 | 0.283 | 0.3331 | 75.73 | 3.77 | 36.2 |
| | Ours (1-Ref) | 0.609 | 0.743 | 0.712 | 0.3647 | 75.22 | 3.84 | 38.3 |

Table 18: Comparison with InterLCM (Li et al., 2025) on FFHQ-Ref Severe.

| Method | #REF | IDS | FaceNet | IDS(REF) | LPIPS↓ | MUSIQ | NIQE↓ | FID↓ |
|---|---|---|---|---|---|---|---|---|
| InterLCM | N/A | 0.266 | 0.643 | 0.190 | 0.3998 | 75.62 | 3.80 | 55.1 |
| Ours | 1 | 0.609 | 0.743 | 0.712 | 0.3647 | 75.22 | 3.84 | 38.3 |

Table 19: Ablation study on the classifier-free guidance scale parameters with FFHQ-Ref Severe.

| $s_i$ | $s_c$ | FFHQ-Ref Severe | | | | | | |
|---|---|---|---|---|---|---|---|---|
| | | IDS | FaceNet | IDS(REF) | LPIPS | MUSIQ | NIQE | FID |
| 1.0 | 1.0 | $0.599 \pm 0.089$ | $0.738 \pm 0.049$ | $0.694 \pm 0.070$ | $0.3645 \pm 0.0723$ | $74.73 \pm 2.73$ | $3.97 \pm 0.61$ | 39.1 |
| 1.0 | 1.2 | $0.608 \pm 0.088$ | $0.742 \pm 0.048$ | $0.719 \pm 0.065$ | $0.3678 \pm 0.0723$ | $75.13 \pm 2.57$ | $3.94 \pm 0.63$ | 38.8 |
| 1.2 | 1.0 | $0.598 \pm 0.091$ | $0.738 \pm 0.050$ | $0.685 \pm 0.073$ | $0.3642 \pm 0.0724$ | $74.84 \pm 2.65$ | $3.85 \pm 0.63$ | 38.8 |
| 1.2 | 1.2 | $0.609 \pm 0.089$ | $0.743 \pm 0.048$ | $0.712 \pm 0.068$ | $0.3647 \pm 0.0722$ | $75.22 \pm 2.46$ | $3.84 \pm 0.64$ | 38.3 |

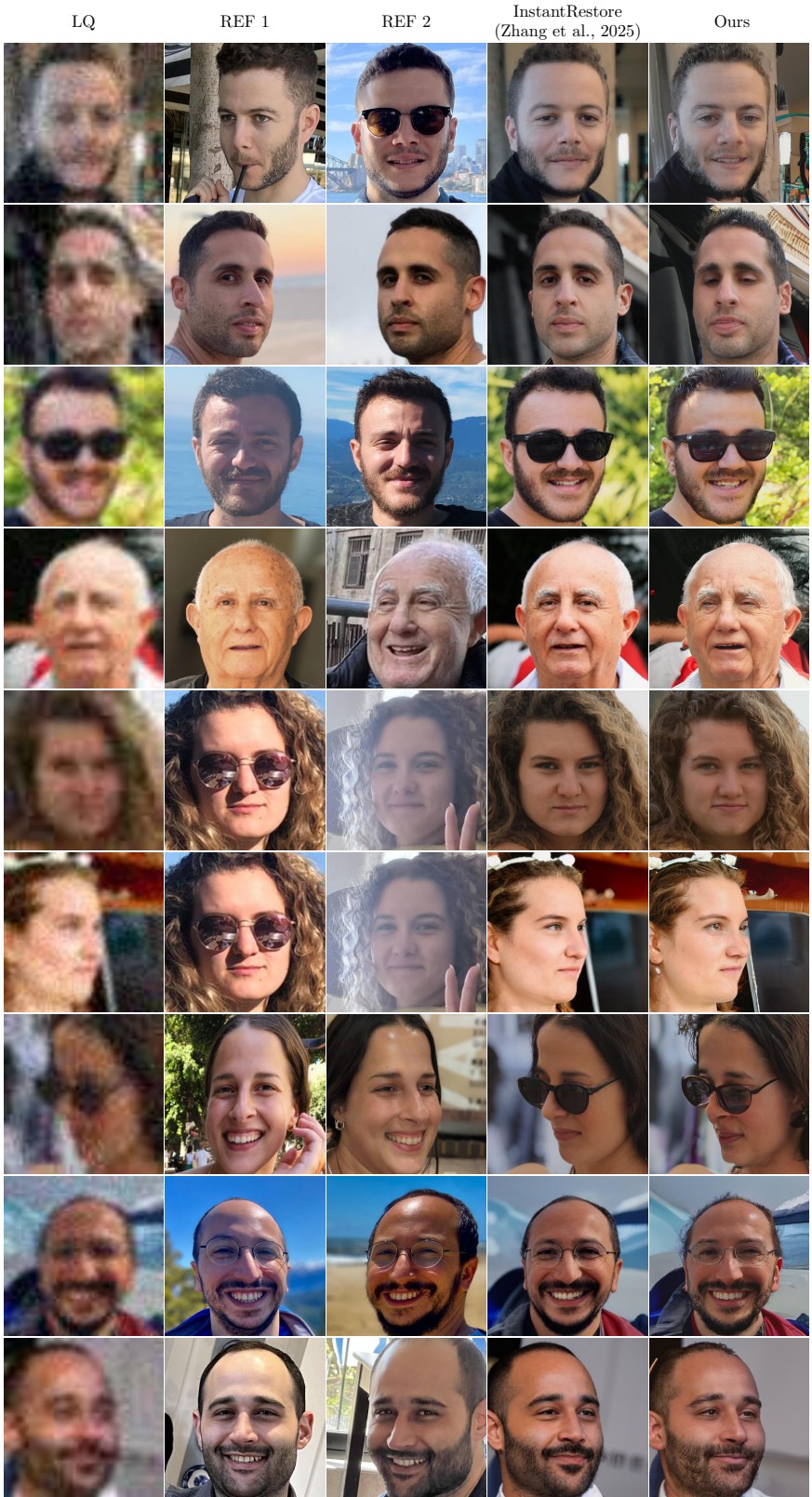

Figure 7: Comparison with InstantRestore on a small set of images from their arXiv preprint source. Our results have better image quality. Zoom in for details.

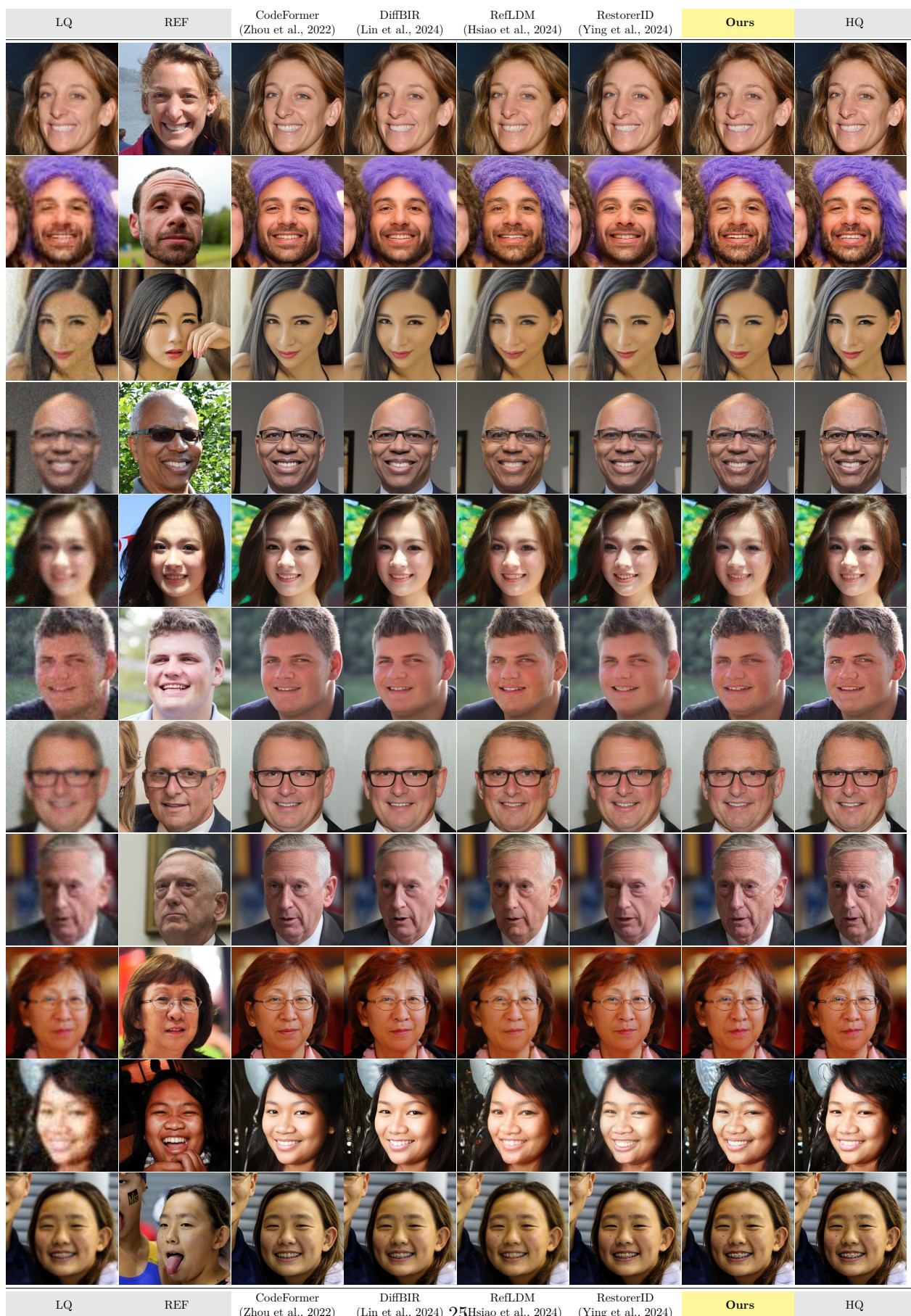

Figure 8: Additional qualitative comparison with other state-of-the-art face restoration methods on FFHQ-Ref Moderate test set. The "REF" column is the high-quality reference face image.

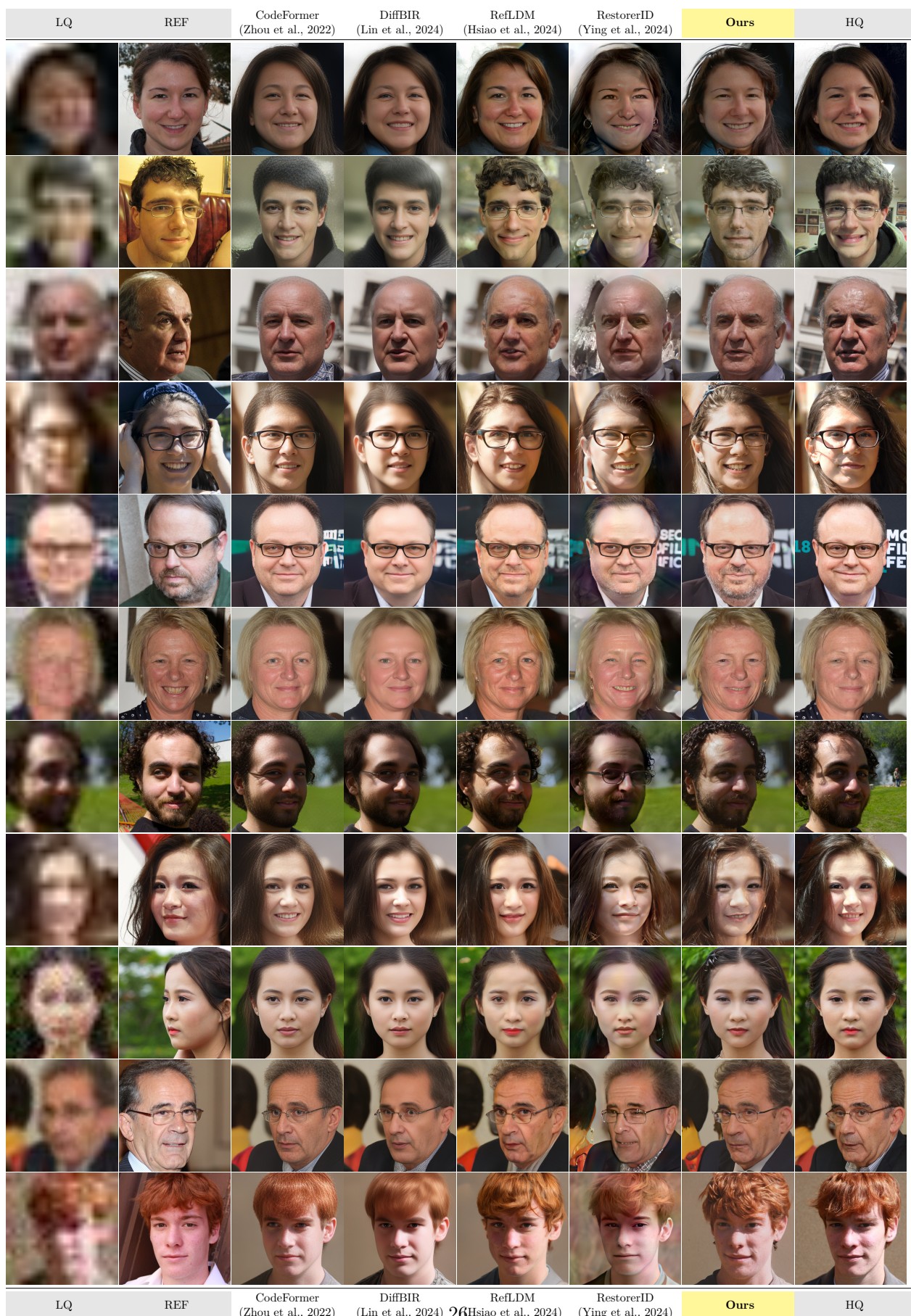

Figure 9: Additional qualitative comparison with other state-of-the-art face restoration methods on FFHQ-Ref Severe test set. The "REF" column is the high-quality reference face image.

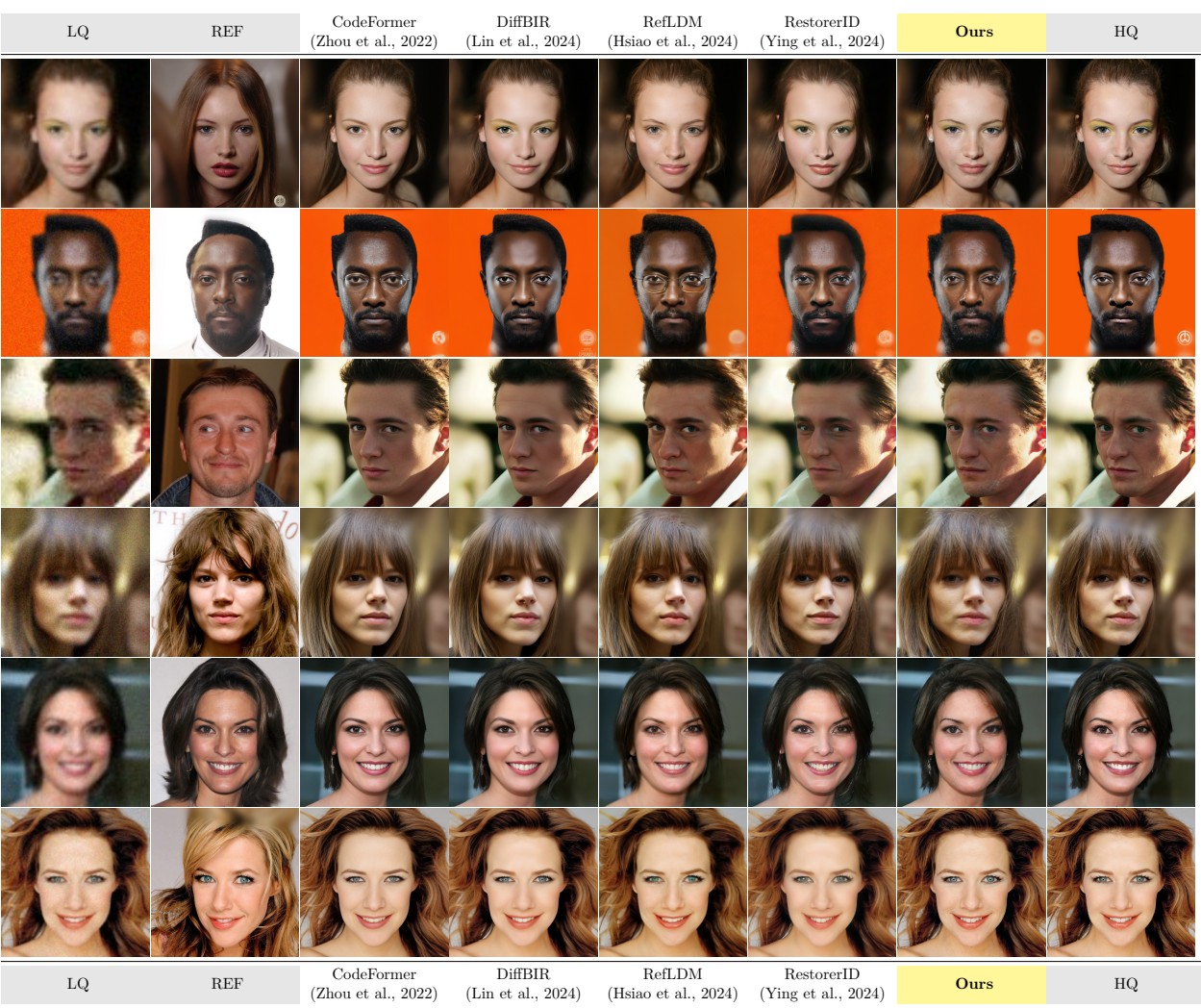

Figure 10: Additional qualitative comparison with other state-of-the-art face restoration methods on CelebA-Ref-Test test set. The "REF" column is the high-quality reference face image.

LQ     REF     No ArcFace No FaRL     No FaRL     No ArcFace     Ours     HQ

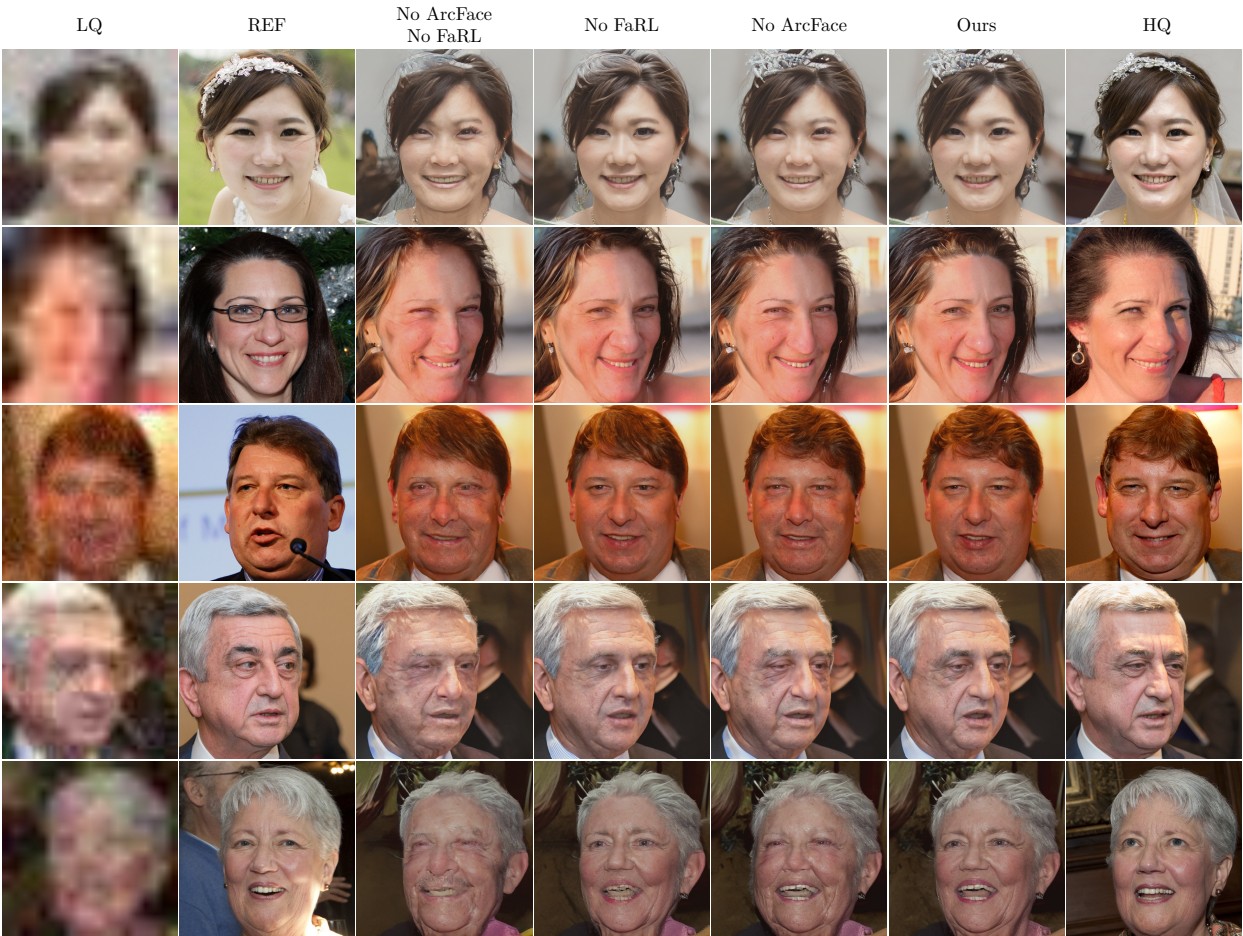

Figure 11: Qualitative ablation on the individual components in our proposed "Composite Context" module. To ease the comparison, we use the same samples shown in the manuscript.

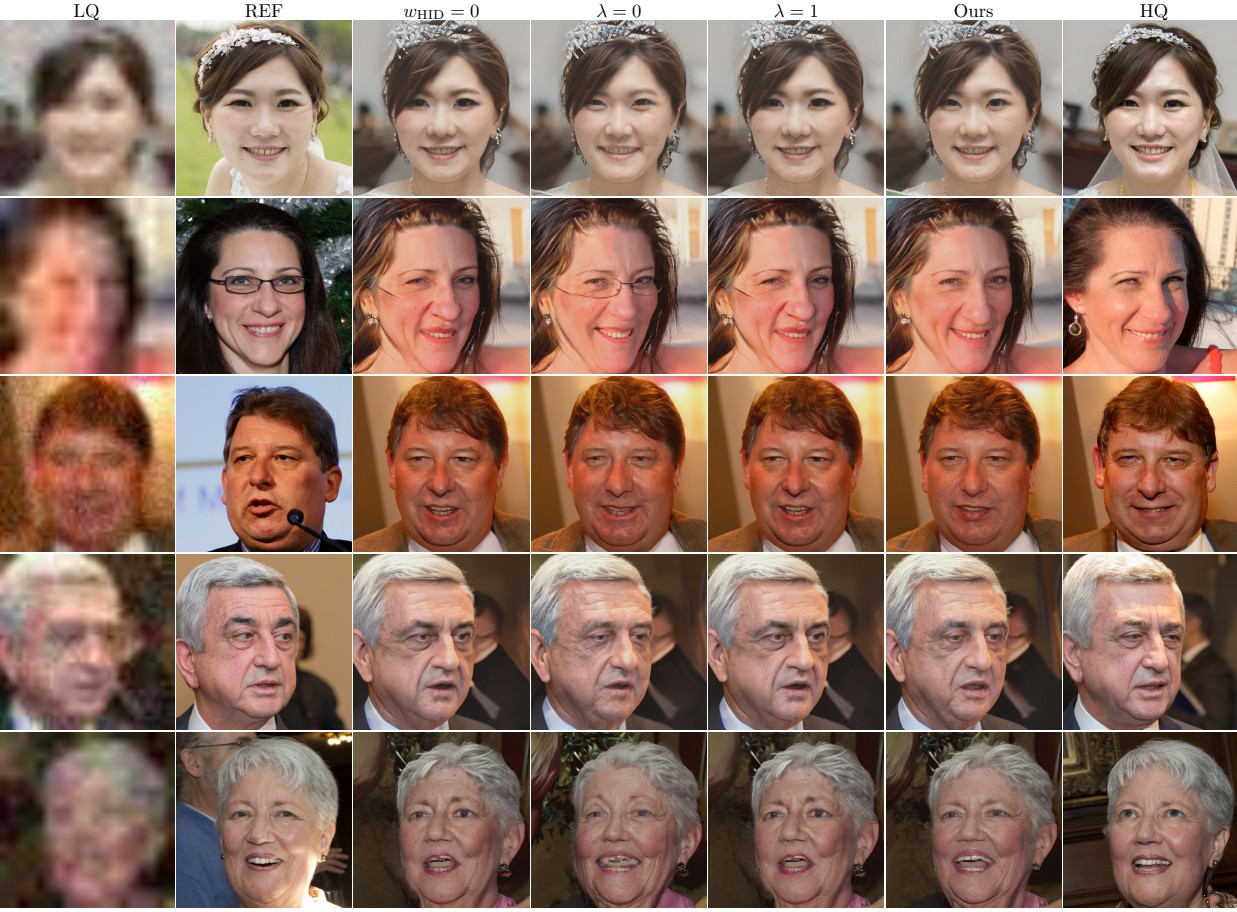

Figure 12: Qualitative ablation on the individual components in our proposed "Hard Example Identity Loss" module. To ease the comparison, we use the same samples shown in the manuscript. The case where $w_{\text{HID}} = 0$ means the whole hard example identity loss has been removed. When $\lambda = 0$, the loss fully relies on the HQ image. When $\lambda = 1$, the loss fully relies on the reference face image.

| LQ | #REF=1 | #REF=5 | HQ |
| --- | --- | --- | --- |

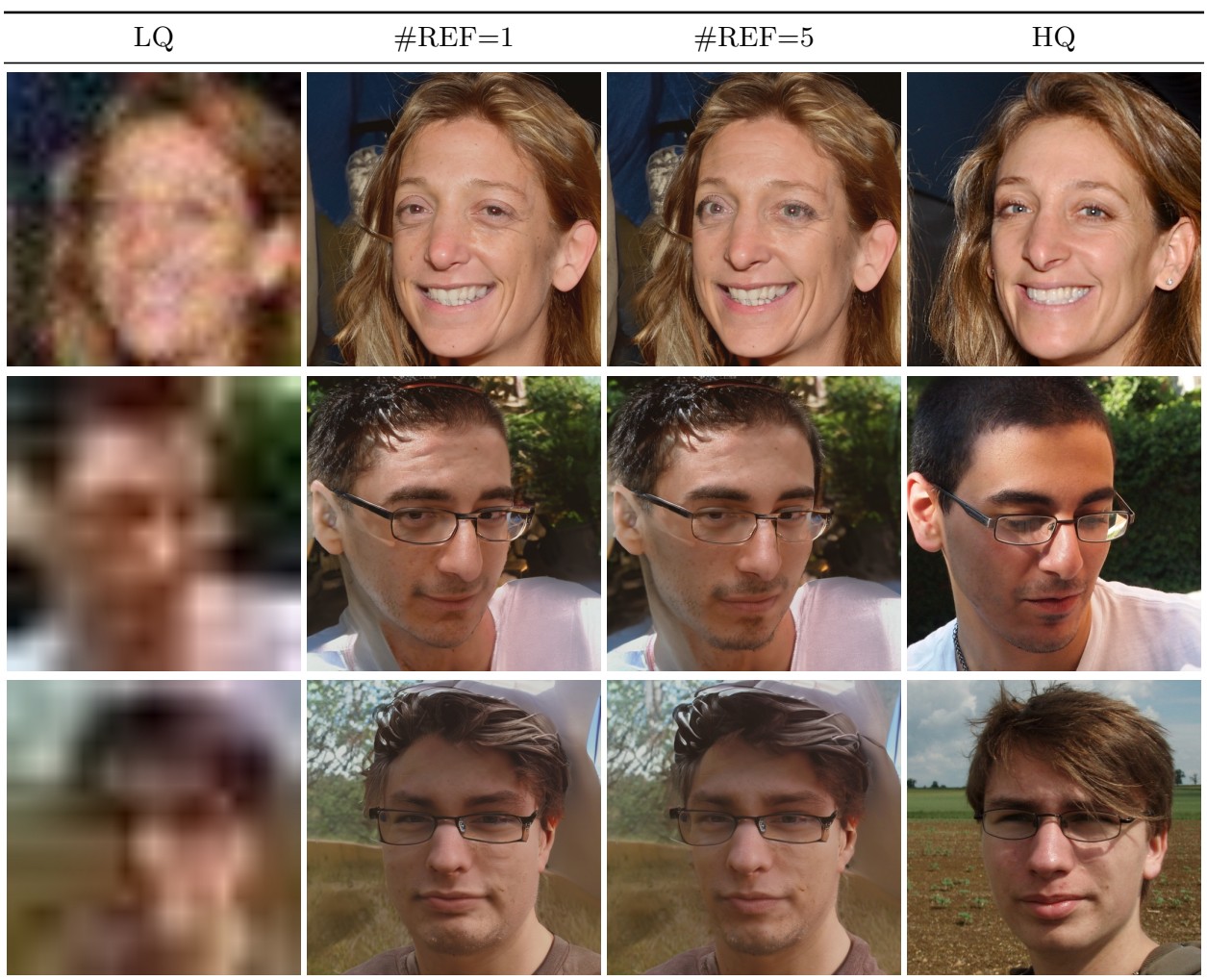

Figure 13: Visualization of multi-reference face restoration on FFHQ-Ref Severe.

| LQ | REF | Result | HQ |
|---|---|---|---|

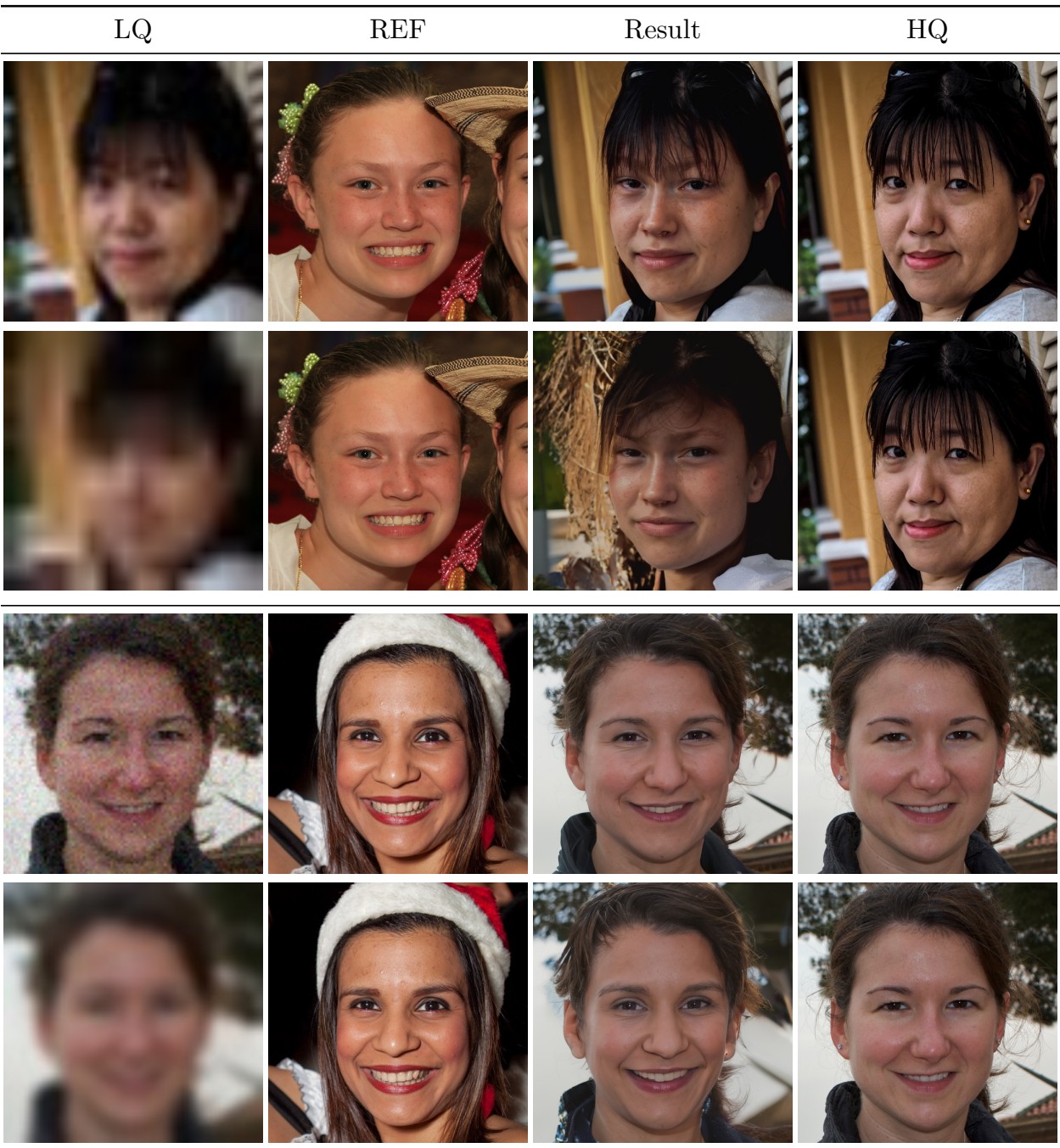

Figure 14: Additional visualizations with wrong reference face. This table is a continuation of the Figure 5 in the manuscript. As discussed in the manuscript, a wrong reference face will leads to some "identity blending" effect depending on how much information is lost from the low-quality input face.

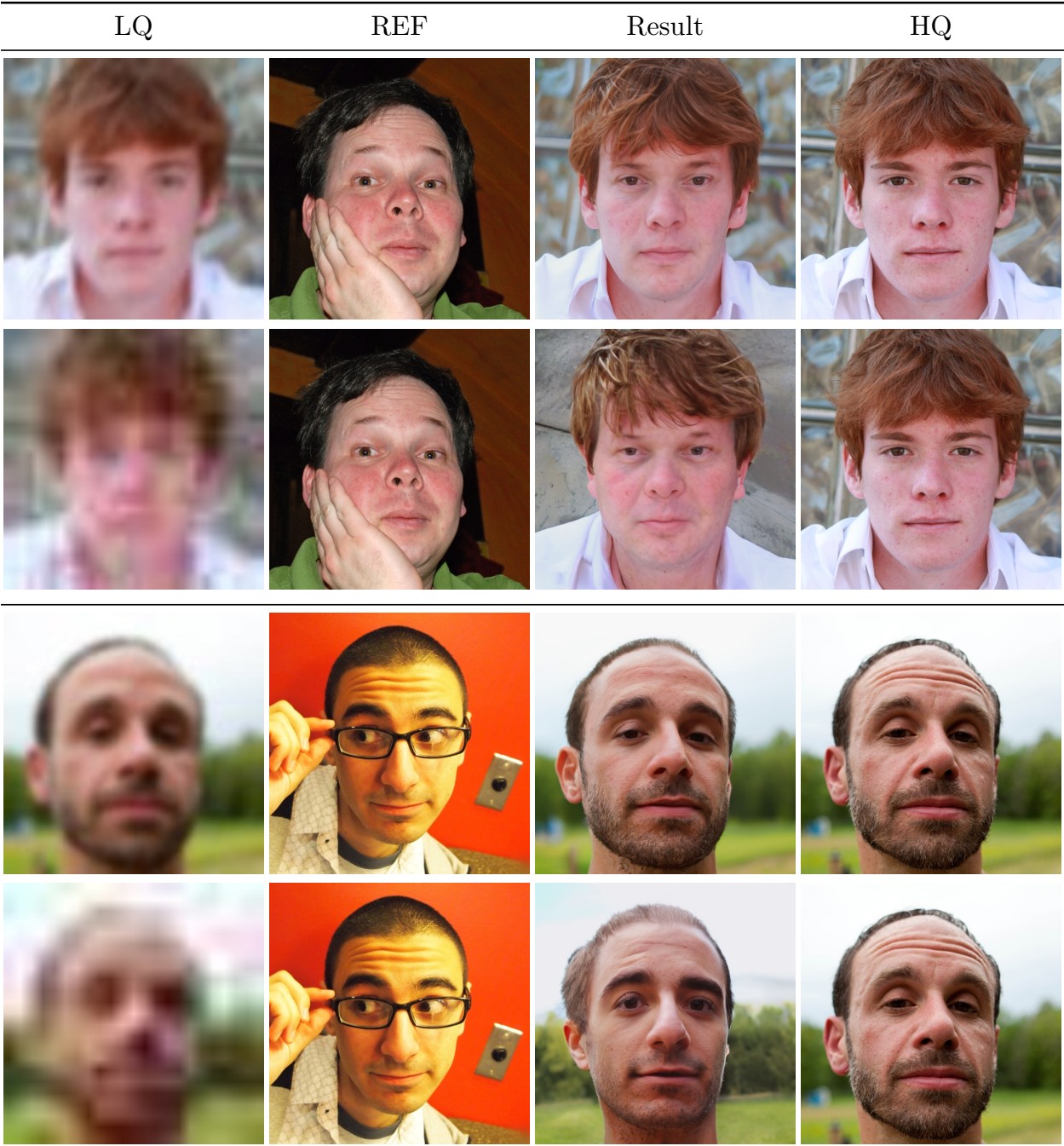

Figure 15: Additional visualizations with wrong reference face. This table is a continuation of the Figure 5 in the manuscript. As discussed in the manuscript, a wrong reference face will leads to some "identity blending" effect depending on how much information is lost from the low-quality input face.

| LQ | REF | Result | HQ |
|----|-----|--------|-----|

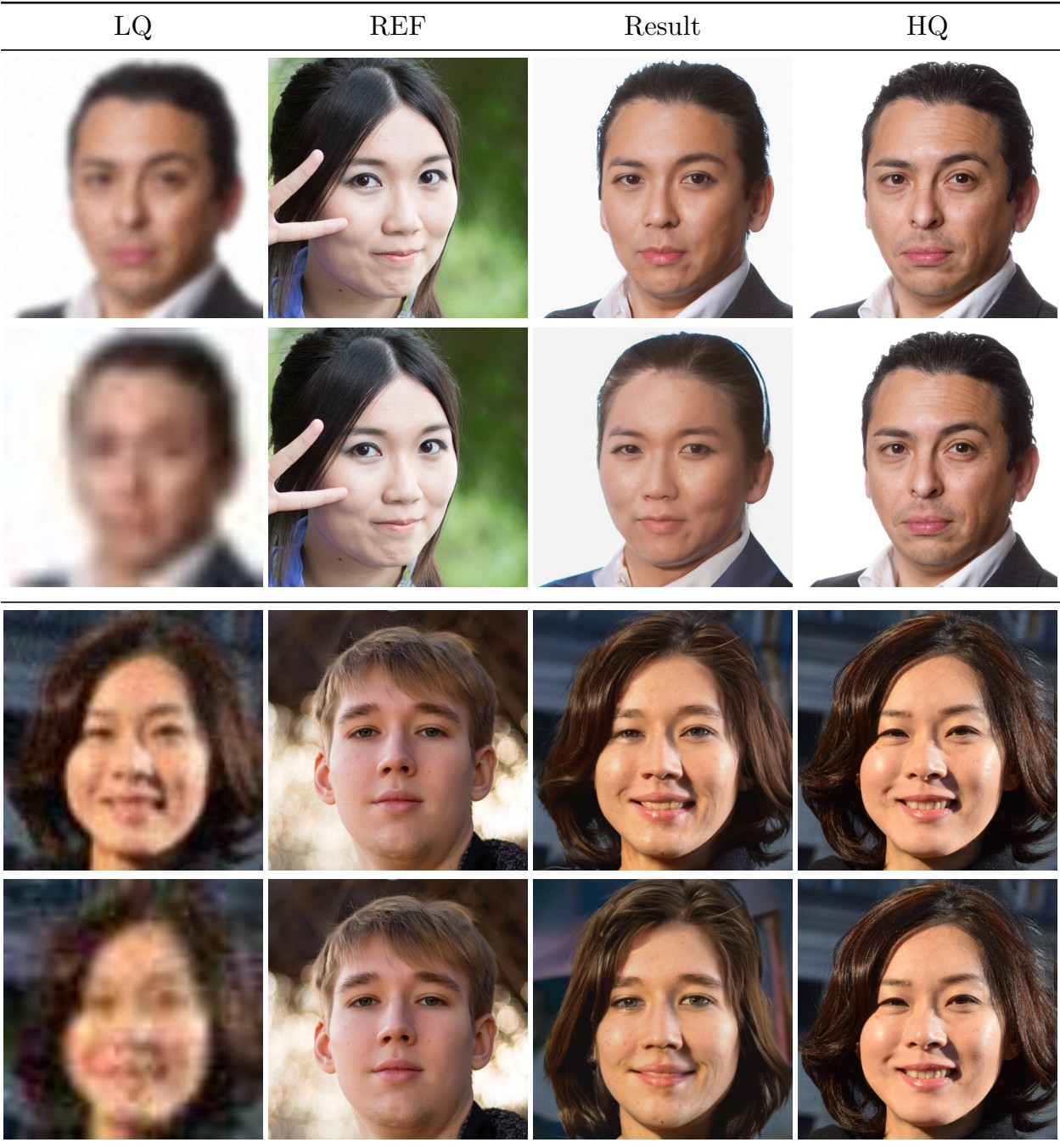

Figure 16: Additional visualizations with wrong reference face. This table is a continuation of the Figure 5 in the manuscript. As discussed in the manuscript, a wrong reference face will leads to some "identity blending" effect depending on how much information is lost from the low-quality input face.

