# OpenReview forum: "Reference-Guided Identity Preserving Face Restoration"
_TMLR — Accepted by TMLR_

### Review · Reviewer_ExB8 · 2026-02-06

**Summary Of Contributions:**

The paper proposes two techniques for identity-preserving face restoration problems. On the one hand, they use multiple feature embeddings trained from a reference face image, which they call composite context. On the other hand, they propose incorporating a hard example into the loss function (using the ground-truth face image), in order to prevent the loss to plateau early at small values.

The paper is very well written, all main ideas are clearly explained and set into context with prior work. This makes the paper accessible also for readers from other research areas. Figure 1 gives an excellent overview of the proposed method.

**Strengths:**

* As mentioned above, the overall presentation is of high quality. The empirical evaluation is extensive, and abaltion studies on the impact of both contributions individually are provided.

* Given the nature of the problem, evaluation of the methods relies heavily on visual inspection. Overall, the paper provides empirical evidence to show the benefits of the proposed modelling technique (see caveats below). Given that I am not an expert on the field, I cannot say whether more recent baselines (all baselines are pre 2025) should be evaluated as well. For example, the InstantRestore method that is mentioned in the paper now has public source code on Github, and thus could be added fully to the comparison.

**Weaknesses:**

* The two main contributions seem closely related to ideas from adjacent fields, or simply extending previous ideas with more powerful representations. This limits the fundamental novelty in terms of methodology; nontheless, the paper shows how to make these ideas work for the problem of interest, which should still be a valuable contribution.

* In some images, the proposed method generates artefacts (e.g. Figure 3, row 3 and 4), or is visibly on par or even slightly inferior to CodeFormer (e.g. Fig 3), which does not use reference images at all. The image comparison with InstantRestore (Fig. 6) also is not in favor of the proposed method. Looking at the overall poor image quality when using multiple reference images (Fig 12), the results in this paper are not fully convincing that the use of reference images can lead to strong improvements.

**Audience:**

Yes

**Audience Explanation:**

The paper is clearly interesting to machine learning practicioners and researchers working on face restoration.

**Broader Impact Concerns:**

The topic of face restoration raises obvious ethical concerns, given its potential mis-use. As I am not an expert in the field, I am not sure whether the Impact Statement could be more specific. For example, it seems that the loss formulation allows to train models that can "restore" low-quality faces with fake reference images.

**Claims And Evidence:**

No

**Claims Explanation:**

The experiments seem to be a fair comparison and are clear, however not fully convincing (see Weaknesses above). I am willing to change this assessment in case the authors can provide additional evidence in favor of their method, or are open to revising their claims.

**Requested Changes:**

* Add additional comparison methods if possible (see above for details)
* Please address the concerns on the visual quality of the restore images discussed above; maybe consider revising your claims.

---

> ### Author Response · Authors · 2026-05-01
> **To Reviewer ExB8**
>
> > In some images, the proposed method generates artefacts (e.g. Figure 3, row 3 and 4), or is visibly on par or even slightly inferior to CodeFormer (e.g. Fig 3), which does not use reference images at all. The image comparison with InstantRestore (Fig. 6) also is not in favor of the proposed method. Looking at the overall poor image quality when using multiple reference images (Fig 12), the results in this paper are not fully convincing that the use of reference images can lead to strong improvements.
> > Please address the concerns on the visual quality of the restore images discussed above; maybe consider revising your claims.
>
> Thank you for pointing this out. It is worth noting that, while in the image restoration literature, image quality metrics are important, when it comes to identity-preserving face restoration applications, identity preservation is often prioritized over the image quality For example, consider the scenario of restoring a picture of a memorable moment (e.g. graduation, wedding etc.) – the restoration result is no longer meaningful if the person's identity changed.
>
> While methods like CodeFormer are preferred by image quality metrics and can indeed generate faces with good texture, their ability to faithfully preserve the facial identity is still lacking. In the scenario of, for example, restoring an old wedding photo, “marrying somebody else” as a result of the model being unable to preserve identity, is worse than “still looking somewhat blurry” in the restoration result. There is a trade-off between identity preservation and image quality, as discussed in the Ref-LDM paper (See their Table 3 “Ablation results for the timestepscaled identity loss”).
> * For instance, the CodeFormer results in Figure 8 (row number 8 and 10) look less like the same person than the ground-truth face, compared to our result.
> * When the input is severely degraded, the benefit of identity preservation shows up more cleanly, as shown in Figure 4 and Figure 9, where the CodeFormer result look less like the same person as the ground-turth than our result.
>
> In the revised manuscript, we briefly discussed this in the paragraph about evaluation metrics.
>
> > Add additional comparison methods if possible (see above for details)
>
> Thanks for the great suggestion! We checked both OSDFace and InstantRestore, and the code for both of them are available as of April 21th, 2026. They are evaluated on the FFHQ-Ref Moderate and FFHQ-Ref Severe datasets, and the results are available below. These results are added in the appendix of the revised version.
>
> ### (1) OSDFace
>
> | Dataset | Method | IDS↑ | FaceNet↑ | IDS(REF)↑ | LPIPS↓ | MUSIQ↑ | NIQE↓ | FID↓ |
> | :--- | :--- | :--- | :--- | :--- | :--- | :--- | :--- | :--- |
> | **FFHQ-Ref Moderate** | OSDFace | 0.830 | 0.839 | 0.577 | 0.2147 | 75.91 | 3.79 | 26.7 |
> | | Ours (1-Ref) | 0.843 | 0.850 | 0.732 | 0.2054 | 75.29 | 3.96 | 25.5 |
> | **FFHQ-Ref Severe** | OSDFace | 0.394 | 0.679 | 0.283 | 0.3331 | 75.73 | 3.77 | 36.2 |
> | | Ours (1-Ref) | 0.609 | 0.743 | 0.712 | 0.3647 | 75.22 | 3.84 | 38.3 |
>
>
> Compard with our method, OSDFace can indeed achieve good image quality as reflected by most image quality metrics. However, its identity preservation falls behind our method especially on the FFHQ-Ref Severe dataset.
>
> ### (2) InstantRestore
>
> | Dataset | Method | IDS↑ | FaceNet↑ | IDS(REF)↑ | LPIPS↓ | MUSIQ↑ | NIQE↓ | FID↓ |
> | :--- | :--- | :--- | :--- | :--- | :--- | :--- | :--- | :--- |
> | **FFHQ-Ref Moderate** | InstantRestore | 0.852 | 0.845 | 0.664 | 0.2165 | 69.85 | 6.08 | 34.7 |
> | | Ours | 0.843 | 0.850 | 0.732 | 0.2054 | 75.29 | 3.96 | 25.5 |
> | **FFHQ-Ref Severe** | InstantRestore | 0.620 | 0.739 | 0.623 | 0.3296 | 70.09 | 6.35 | 49.1 |
> | | Ours | 0.609 | 0.743 | 0.712 | 0.3647 | 75.22 | 3.84 | 38.3 |
>
> According to the results, while InstantRestore indeed achieve a slightly higher IDS as Arcface is used in its training procedure, its FaceNet metric and IDS(REF) metric are surpassed by our method, especially by a large margin in IDS(REF). In particular, as both ground-truth image (used by IDS) and the reference image (used by IDS(REF)) are the same person, a better method should achieve high performance in both IDS and IDS(REF). That suggests that InstantRestore generates faces that look like the same person as the ground-truth face, but not necessarily the same person as the reference face (even though they are the same person). In contrast, our method achieves higher IDS(REF) than InstantRestore by a large margin. Besides, our method outperforms InstantRestore in image quality, as almost all quality metrics of our method are better than those of InstantRestore on FFHQ-Ref Moderate and FFHQ-Ref Severe.

---

### Review · Reviewer_qSmu · 2026-03-05

**Summary Of Contributions:**

This paper studies the accurate reconstruction of images from a low-quality image. They focus on face restoration and assume access to a high-quality image of the same person.
The main idea is to train a conditional latent diffusion model which takes a (pretrained) embedding of the high quality image as context. The paper suggests to use an identity loss on the reconstructions (for arbitrary times) which encourages similarity of the embeddings of the reconstruction and either the low quality image or the high quality reference image.
They use classifier free guidance which allows them to also consider multiple reference images (by averaging the denoisers).

Strengths:
- Even without a background in the field, I could follow the paper reasonably well.
- The chosen approach seems reasonable.
- Results are OK

Weaknesses:
- The contribution appears a bit incremental and results are only a bit better (this is not an acceptance criterion)

**Additional Comments:**

- To what extent are the shown examples cherry-picked?

**Audience:**

Yes

**Audience Explanation:**

People interested in this problem would be interested in this work. The paper is quite accesible and therefore of interest to the broader cv community.

**Broader Impact Concerns:**

While the considered problem might have some ethical implications, those are probably not specific for this method.

**Claims And Evidence:**

Yes

**Claims Explanation:**

- The method is compared to several baselines using several different metrics.
- Several visual examples

Few minor questions below

**Requested Changes:**

- It would be good to report some error estimates of the evaluation to assess significance (in the appendix those are contained and often show no significant difference)
- This is more  a questions: As also briefly discussed in the submission it is not clear whether ID is the right metric to measure reconstruction, however, there is probably no better alternative. Including ID(REF) with and uparrow appears to be a bit strange to me. This could be maximized by outputting the high-quality image which is not a good reconstruction.
- Some parts of the appendix seem to be copied from a rebuttal of a prior submission ("answer to the previous question"). Please proofread carefully and potentially improve the structure of the appendix.
- Please check again whether code for omitted baselines is still not available.

---

> ### Author Response · Authors · 2026-05-01
> **To Reviewer qSmu (1/2)**
>
> > The contribution appears a bit incremental and results are only a bit better (this is not an acceptance criterion)
>
> Thank you for the comment. As summarized in Appendix A.2 (in the revised version), our method differs from the related works in: (1) it leverages an ensemble representation instead of partial information through face feature. (2) We address the overlooked learning inefficiency issue in identity loss. (3) We alleviate the model’s reliance on multi-ref training data for better scalability.
>
> > It would be good to report some error estimates of the evaluation to assess significance (in the appendix those are contained and often show no significant difference)
>
> Thank you for the suggestion. For the brevity of the manuscript, we mentioned that “More details including standard deviation are available in Appendix” in the caption of Table 1 and Table 2.
>
> > As also briefly discussed in the submission it is not clear whether ID is the right metric to measure reconstruction, however, there is probably no better alternative.
>
> We agree with the reviewer. To the best of our knowledge, there is not yet a better method for measuring “whether the person identity is well-preserved in the restored face image” than IDS and similar methods (like FaceNet metric used in the paper, which is essentially a different face embedding model but follows a similar mechanism). That is the reason why we chose to provide two different identity metrics (IDS and FaceNet). Meanwhile, IDS is a common choice among related works, including Ref-LDM.
>
> > Including ID(REF) with and uparrow appears to be a bit strange to me. This could be maximized by outputting the high-quality image which is not a good reconstruction.
>
> We agree with the reviewer. If the model learns a behavior to “copy” the high-quality reference face, it can potentially cheat the IDS but this is not a good restoration. However, note that the model is trained with the reconstruction loss against the ground-truth, where “copying” the reference face instead leads to a high reconstruction loss penalty.
>
> On the other hand, neither IDS and IDS(REF) can be maximized through outputting a high-quality image. IDS is calculated as the cosine similarity between the face embedding vectors of the ground-truth and the generated face image, while the IDS(REF) is that between the reference face and the generated face image. The face embedding is designed to represent person identity, instead of image quality. Thus, IDS and similar metrics are measuring identity-preserving performance (the main goal of this paper), instead of face image quality.
>
> > Please proofread carefully and potentially improve the structure of the appendix.
>
> Thank you for the suggestion. We have carefully proofread the appendix and reorganized its structure. For ease of reading and understanding the structure of the appendix, we also added a table of contents before the appendix sections.
>
> > To what extent are the shown examples cherry-picked?
>
> The visualization samples are cherry-picked to an extent where none of these methods (all other methods and our methods) is generating clearly problematic images, such as those with strange face structure or texture.

---

> ### Author Response · Authors · 2026-05-01
> **To Reviewer qSmu (2/2)**
>
> > Please check again whether code for omitted baselines is still not available.
>
> Thanks for the great suggestion! We checked both OSDFace and InstantRestore, and the code for both of them are available as of April 21th, 2026. They are evaluated on the FFHQ-Ref Moderate and FFHQ-Ref Severe datasets, and the results are available below. These results are added in the appendix of the revised version.
>
> ## (1) OSDFace
>
> | Dataset | Method | IDS↑ | FaceNet↑ | IDS(REF)↑ | LPIPS↓ | MUSIQ↑ | NIQE↓ | FID↓ |
> | :--- | :--- | :--- | :--- | :--- | :--- | :--- | :--- | :--- |
> | **FFHQ-Ref Moderate** | OSDFace | 0.830 | 0.839 | 0.577 | 0.2147 | 75.91 | 3.79 | 26.7 |
> | | Ours (1-Ref) | 0.843 | 0.850 | 0.732 | 0.2054 | 75.29 | 3.96 | 25.5 |
> | **FFHQ-Ref Severe** | OSDFace | 0.394 | 0.679 | 0.283 | 0.3331 | 75.73 | 3.77 | 36.2 |
> | | Ours (1-Ref) | 0.609 | 0.743 | 0.712 | 0.3647 | 75.22 | 3.84 | 38.3 |
>
>
> Compard with our method, OSDFace can indeed achieve good image quality as reflected by most image quality metrics. However, its identity preservation falls behind our method especially on the FFHQ-Ref Severe dataset.
>
> ## (2) InstantRestore
>
> | Dataset | Method | IDS↑ | FaceNet↑ | IDS(REF)↑ | LPIPS↓ | MUSIQ↑ | NIQE↓ | FID↓ |
> | :--- | :--- | :--- | :--- | :--- | :--- | :--- | :--- | :--- |
> | **FFHQ-Ref Moderate** | InstantRestore | 0.852 | 0.845 | 0.664 | 0.2165 | 69.85 | 6.08 | 34.7 |
> | | Ours | 0.843 | 0.850 | 0.732 | 0.2054 | 75.29 | 3.96 | 25.5 |
> | **FFHQ-Ref Severe** | InstantRestore | 0.620 | 0.739 | 0.623 | 0.3296 | 70.09 | 6.35 | 49.1 |
> | | Ours | 0.609 | 0.743 | 0.712 | 0.3647 | 75.22 | 3.84 | 38.3 |
>
> According to the results, while InstantRestore indeed achieve a slightly higher IDS as Arcface is used in its training procedure, its FaceNet metric and IDS(REF) metric are surpassed by our method, especially by a large margin in IDS(REF). In particular, as both ground-truth image (used by IDS) and the reference image (used by IDS(REF)) are the same person, a better method should achieve high performance in both IDS and IDS(REF). That suggests that InstantRestore generates faces that look like the same person as the ground-truth face, but not necessarily the same person as the reference face (even though they are the same person). In contrast, our method achieves higher IDS(REF) than InstantRestore by a large margin. Besides, our method outperforms InstantRestore in image quality, as almost all quality metrics of our method are better than those of InstantRestore on FFHQ-Ref Moderate and FFHQ-Ref Severe.

---

### Review · Reviewer_rTdj · 2026-04-09

**Summary Of Contributions:**

I apologize for the late submission of this review.

I enjoyed in reading this manuscript. The paper uses latent diffusion framework for restoring low-quality images to high-quality images, by conditioning on the representational ensemble of identity representation and general representation. the identity representation is from a pre-trained metric learning model, ArcFace. The architecture appears sound to me.

**Additional Comments:**

NA

**Audience:**

Yes

**Audience Explanation:**

Yes, researchers in the image restoration community would likely be interested in this work.

**Claims And Evidence:**

Yes

**Claims Explanation:**

Yes, the claims made in submission supported by evidence.

**Requested Changes:**

The experimental results have been examined by earlier reviewers. I don't have comments on them. I have two concerns.

## 1. Motivation clarification. I read through the introduction, however, I didn't see the authors clearly present the motivations.
eg. in paragraph 2:
> "Nevertheless, the existing methods do not fully exploit the potential of reference faces, and hence there is room for improvement in both identity preservation and image quality."

and paragraph 3
> P3 "In this paper, to simplify architecture and more effectively utilize the reference face images and further
enhance the performance of reference-based face restoration, we propose two independent modules that
exploit the reference face in two different aspects: representation and supervision."


i would like to suggest to clearly state what are the problems/motivations to solve and what would be niche for this work.


## 2 a bit overclaim regarding regarding "comprehensive representation"
> P4: "comprehensive representation" do you mean ensemble of representations?

This word "comprehensive" seems a bit overclaim. It's only the ensemble of representations to conditioning the latent diffusion model.

I'm looking forward to reading revised manuscript.

---

> ### Comment · Reviewer_rTdj · 2026-04-09
> **One more request.**
>
> I also checked the bibliography, could the authors check the accuracy of the bibliography? For example:
>
> > Ashish Vaswani, Noam Shazeer, Niki Parmar, Jakob Uszkoreit, Llion Jones, Aidan N. Gomez, Lukasz Kaiser, and Illia
> Polosukhin. Attention is all you need, 2023. URL https://arxiv.org/abs/1706.03762.

---

> ### Author Response · Authors · 2026-05-01
> **To Reviewer rTdj**
>
> > 1. Motivation clarification. I read through the introduction, however, I didn't see the authors clearly present the motivations.
>
> Thanks for the suggestion. We added more explanations in the introduction section as suggested for better clarity.
> * In paragraph 2, we revised the sentence as follows: Nevertheless, the existing methods do not fully exploit the potential of reference faces; only partial representation of the reference face is used, and not sufficiently involved in supervision, and hence there is room for improvement in both identity preservation and image quality.
> * In paragraph 3, we appended this at the end of this paragraph: “Respectively, we propose Composite Context for the representation aspect, and Hard Example Identity Loss for the supervision aspect. They will be explained in the following text.”
> * In paragraph 4 (for Composite Context), we added: “It is designed in order to avoid using only a partial representation of the reference face, as related works
> only use either the high-level information (such as person identity), or the low-level appearance information (such as skin texture).\footnote{The details are discussed in Section 3.1}”
> * In paragraph 5 (for Hard Example Identity Loss), we added: “It is designed in order to avoid the learning inefficiency issue with the existing loss functions.\footnote{See Section 3.2 for detailed explanations including the loss curves.}”
>
> > 2. a bit overclaim regarding regarding "comprehensive representation"
>
> Thanks for the suggestion. We revised the manuscript, replaced “comprehensive representation” into “ensembled representation” for better clarity.
>
> > 3. could the authors check the accuracy of the bibliography?
>
> Thanks for the suggestion. We checked the references and replaced all the arxiv preprint papers where possible. Some references such as “Classifier-free diffusion guidance” are unpublished so we could not replace the arxiv reference.

---

### Decision · Action_Editor_vPxV · 2026-05-11

**Recommendation:** Accept with minor revision

**Additional Comments:**

I'm recommending that the paper be accepted, conditioned on it being revised in a few places. In general, the empirical evidence does not support the statement (page 2)

> Though simple, our method effectively and consistently outperforms the previous methods in face identity preservation.

I can accept the paper if such wording is revised to better match the empirical findings.

**Audience:**

Yes

**Audience Explanation:**

The paper is concerned with image restoration of face images with the constraint of preserving identity. This is an established task within the image restoration community, and I expect them to find the work interesting.

**Claims And Evidence:**

Yes

**Claims Explanation:**

The answer is only a partial yes, and corrections will be needed.

The empirical investigations do a good job of shedding light on the proposed method. However, the paper does state that (page 2)

> Though simple, our method effectively and consistently outperforms the previous methods in face identity preservation.

which is not backed up by the empirical evidence. The authors will need to revise such statements.